# ORP1L mediated PI(4)P signaling at ER-lysosome-mitochondrion three-way contact contributes to mitochondrial division

Maxime Boutry[1] & Peter K. Kim [1,2 ✉]

Mitochondrial division is not an autonomous event but involves multiple organelles, including the endoplasmic reticulum (ER) and lysosomes. Whereas the ER drives the constriction of mitochondrial membranes, the role of lysosomes in mitochondrial division is not known. Here, using super-resolution live-cell imaging, we investigate the recruitment of lysosomes to the site of mitochondrial division. We find that the ER recruits lysosomes to the site of division through the interaction of VAMP-associated proteins (VAPs) with the lysosomal lipid transfer protein ORP1L to induce a three-way contact between the ER, lysosome, and the mitochondrion. We also show that ORP1L might transport phosphatidylinositol-4-phosphate (PI(4)P) from lysosomes to mitochondria, as inhibiting its transfer or depleting PI(4)P at the mitochondrial division site impairs fission, demonstrating a direct role for PI(4)P in the division process. Our findings support a model where the ER recruits lysosomes to act in concert at the fission site for the efficient division of mitochondria.

[1] Cell Biology Program, Hospital for Sick Children, Peter Gilgan Centre for Research and Learning, Toronto, ON, Canada. [2] Department of Biochemistry, University of Toronto, Toronto, ON, Canada. ✉email: pkim@sickkids.ca

Mitochondria are essential organelles whose function is required for cellular homeostasis and physiology[1,2]. Mitochondrial functions are intimately linked to their shape, that is maintained by a tightly regulated balance of fusion and division (fission) events[3]. Numerous diseases are caused by mutations in genes encoding mitochondrial fusion and fission factors[4] stressing the need to fully elucidate the mechanisms that mediate and regulate mitochondrial dynamics.

Recent advances in our understanding of mitochondrial dynamics suggest that the cooperation of multiple organelles mediate mitochondrial division by establishing membrane contact sites with mitochondria[5–7]. The division of mitochondria is initiated at contact sites with the endoplasmic reticulum (ER) where the ER drives the initial constriction of mitochondrial membranes[8]. This allows adapters such as MFF (Mitochondrial Fission Factor) to recruit the large guanosine triphosphatase (GTPase) Dynamin-related protein-1 (Drp1) which further constricts mitochondria[9]. The late steps of division, culminating in the fission of mitochondrial membranes, are not fully understood and are still debated[4,10]. Recently, late endosomes/lysosomes (hereafter referred to as lysosomes) and Golgi-derived vesicles were reported to participate in the mitochondrial division process[6,7]. Both of these organelles are found localized at the site of division where they are believed to form membrane contact sites with mitochondria. However, their role in mitochondrial division is unclear.

Lysosomes-mitochondria contact sites were shown to require the small GTPase Rab7 (encoded by the *RAB7A* gene), an important regulator of lysosomes functions[7]. Rab7 cycles between an inactive cytosolic GDP bound state and an active GTP bound state that localizes to lysosomes. At the lysosome, GTP bound Rab7 recruits several effectors modulating a myriad of functions[11]. Elegant work by Dimitri Krainc's group demonstrated that GTP bound Rab7 promotes lysosome-mitochondria contacts and that Rab7 GTP hydrolysis mediates the dissociation of the two organelles. Further, they found that preventing lysosomes dissociation from the mitochondrial division site reduced mitochondrial fission[7]. However, it is not known how Rab7 mediates lysosomes recruitment to the site of mitochondrial division.

Rab7 also promotes the formation of ER-lysosome contact sites by binding to a number of effectors such as ORP1L, protrudin, and PDZD8[12–14]. The dual involvement of Rab7 in ER-lysosome and mitochondria-lysosome contacts raises the possibility of a formation of a three-way contact between these organelles during the division process. A three-way contact can be loosely defined as the formation of two different contact sites at close apposition to each other. Such contact was recently described between the ER, lysosome, and mitochondria where an ER-lysosome contact site was shown in close apposition to that of an ER-mitochondria contact site[15]. However, whether a similar contact contributes to mitochondrial fission was not investigated. In a three-way contact model for mitochondrial fission, lysosomes could be recruited to mitochondrial division site via the formation of an ER-lysosome contact. This recruitment of lysosomes by the ER would bring lysosomes in close apposition to the mitochondrial division site and establish a three-way contact between these three organelles during mitochondrial division.

To test this hypothesis, we used quantitative super-resolution microscopy to show that the ER recruits lysosomes through the interaction between VAMP-associated proteins (VAPs) with the lysosomal localized ORP1L at the site of mitochondrial division to form a three-way contact suggesting that the ER regulates mitochondrial dynamics by coordinating the timed assembly of multiple components required for mitochondrial division. Furthermore, rescue experiments using ORP1L knockout cells show that its lipid transfer domain is crucial for mitochondrial division, and depleting phosphatidylinositol-4-phosphate (PI(4)P) either at the site of division or at lysosomes prevents division suggesting the importance of PI(4)P in the division process. Together, our results support a model in which ORP1L is required to recruit lysosomes to the site of mitochondrial division and to provide PI(4)P from lysosomes to the division site during a late step of the mitochondrial division process.

## Results

**The ER, lysosomes, and mitochondria form a three-way contact that participate in mitochondrial division.** To investigate whether the ER and lysosomes act in concert during mitochondrial division via the formation of a three-way contact, we first looked for three-way contacts between mitochondria, lysosomes, and the ER in COS-7 cells at basal state by live-imaging using a sub-Airy pinhole confocal super-resolution microscopy system. An example of a three-way contact between these organelles that shows the ER wrapped around a lysosome in direct contact with a mitochondrion is shown in Fig. 1a. These three-way contacts were observed in multiple cell lines (Fig. 1b) suggesting a commonality within cells. To evaluate the frequency of ER-lysosome-mitochondria three-way contacts, we monitored the presence of the ER at lysosome-mitochondria contacts using live imaging as lysosome-mitochondria contacts were less numerous and easier to detect in cells than ER-mitochondria contacts. Our criterion for a three-way contact was the presence of the ER at a lysosome-mitochondrion contact for at least 10 s. Around 75% of lysosome-mitochondria contacts were marked by the ER across all cell lines examined (Fig. 1b, c) and many of these three-way contacts were stable far beyond the 10 s time point (Supplementary Fig. 1a, b).

To examine whether the three-way contacts participate in mitochondrial division, we imaged HeLa cells expressing mitochondrial, ER, and lysosomal markers for a duration of 2 min and monitored the outcome of 100 randomly selected pre-existing three-way contacts from 20 cells. Most contacts did not lead to division but instead untethered (86/100 contacts) or remained tethered (11/100 contacts). Of the contact events visualized, a small proportion of them resulted in a mitochondrial fission event (3/100 contacts) (Supplementary Fig. 1c) suggesting most three-way contacts are not involved in mitochondrial division. Next, we quantified the fraction of mitochondrial division events with either ER or Lysosomes or both at the site of division before membrane fission. We found that the ER marked all division events observed ($n = 73/73$ events from 30 cells) and lysosomes were present in over half of them ($n = 43/73$ events) (Fig. 1d, e). In ~91% of the division events involving lysosomes, an ER-mitochondria contact preceded lysosome recruitment to the site of division (Fig. 1f). These findings suggest that lysosomes may be recruited to the division site by the ER and contribute to mitochondrial division after the initial constriction of mitochondria by the ER.

To determine at which step of the mitochondrial division process lysosomes are recruited, we compared lysosome recruitment to that of Drp1 and Arf1. Drp1 is recruited to the division site after the initial constriction by the ER to further constrict the mitochondrial membranes[9,16], while Golgi vesicles harboring Arf1 are recruited after Drp1 and participate in the late steps of the division process by an undefined mechanism[6]. We found that in the majority of fission events, lysosomes were recruited to mitochondrial division sites after Drp1 but before the Golgi vesicles (Fig. 1g–l). Taken together, our data suggest that a three-way contact between lysosomes, the ER, and mitochondria forms at the mitochondrial fission site during a late step of the division process after the recruitment of Drp1.

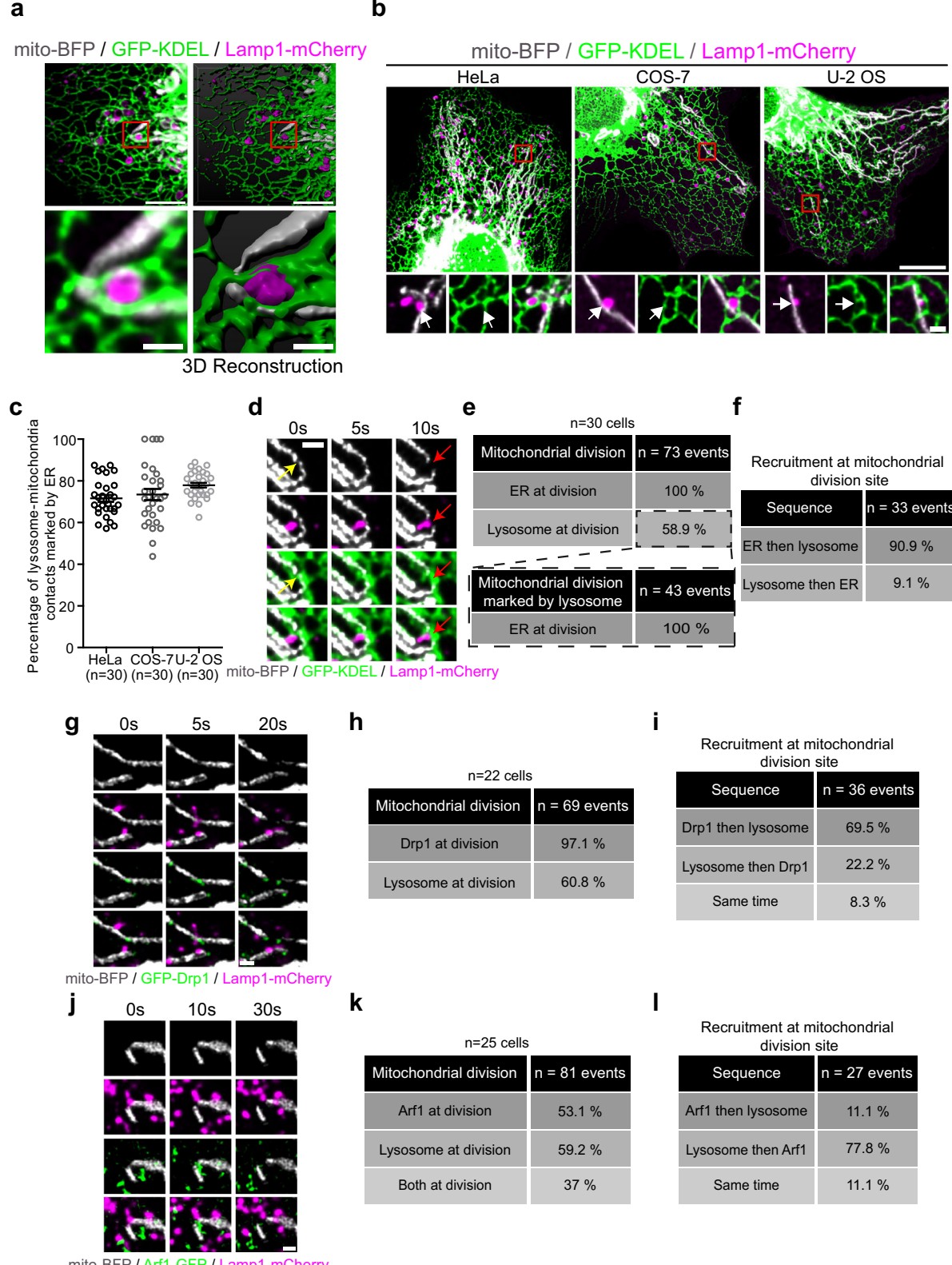

**Rab7 mediates the formation of a subset of three-way contacts between lysosomes, mitochondria, and the ER**. Rab7 has been implicated in lysosome contact sites with the ER and with mitochondria[7,13,17]. More recently, it has been suggested that Rab7 may also participate in the formation of ER-lysosome-mitochondria three-way contacts[15]. Our observations showing lysosome recruitment to the mitochondrial constriction site after

the ER (Fig. 1f) suggest that lysosomes may be recruited there by the ER through the formation of contact sites. We, therefore, reasoned that Rab7 might mediate the contact between lysosomes and the ER at, or near, the ER-mitochondria contact sites to establish the three-way contacts that notably participate in mitochondrial division. To validate this hypothesis and quantify the role of Rab7 in the formation of three-way contacts, we

**Fig. 1 Lysosomes, mitochondria, and the ER are involved in a three-way contact that participates in a late step of mitochondrial division. a** Representative image and three-dimensional reconstruction of an ER-lysosome-mitochondrion three-way contact in a COS-7 cell. Scale bars: 5 and 1 µm (inset). **b** HeLa, COS-7, and U-2 OS expressing mito-BFP, GFP-KDEL, and Lamp1-mCherry. Insets are the magnification of the area indicated by the red box showing a lysosome-mitochondria contacts marked by the ER. White arrow indicates a lysosome-mitochondria contact. Scale bars: 10 µm and 1 µm (inset). **c** Quantification of the percentage of lysosome-mitochondria contacts that are marked by the ER in HeLa, COS-7, and U-2 OS cells. The graph shows the mean ± SEM. Cells from three independent experiments. **d** Live-cell imaging of HeLa cells expressing mito-BFP, GFP-KDEL, and Lamp1-mCherry over 10 s. Yellow arrows indicate an ER-mediated mitochondrial constriction site and red arrows indicate a mitochondrial fission event. Scale bar: 1 µm. **e** Percentage of mitochondrial division events (cells from three independent experiments) marked by the ER or lysosomes and the percentage of lysosomes marked division events that were also marked by the ER in HeLa cells. **f** Sequence of recruitment of the ER and lysosomes at mitochondrial division site for events that were positive for both the ER and lysosome in **e**. **g–i** Live-cell imaging of HeLa cells expressing GFP-Drp1, mito-BFP, and Lamp1-mCherry (**g**). Scale bar: 1 µm. **h** Percentage of mitochondrial division events marked by Drp1 and by lysosomes (cells from two independent experiments). **i** Sequence of recruitment of Drp1 and lysosomes at mitochondrial division site for events that were positive for both Drp1 and lysosome. **j–l** Live-cell imaging of HeLa cells expressing mito-BFP, GFP-Arf1, and Lamp1-mCherry (**j**). Scale bar: 1 µm. **k** Percentage of mitochondrial division events marked by Arf1 and by lysosomes (cells from two independent experiments). **l** Sequence of recruitment of Arf1 and lysosomes at mitochondrial division site for events that were positive for both Arf1 and lysosome. **f**, **i**, **l** Division sites that were already positive for both markers at the beginning of the acquisition were excluded from the analysis.

imaged Rab7, the ER, and mitochondria and found that Rab7 positive lysosomes were present at ER-lysosome-mitochondria contacts (Fig. 2a). Further, overexpressing Rab7 slightly but significantly increased the proportion of Lamp1 positive structures found with mitochondria and the ER as compared to that of a control vector (Fig. 2b) without any increase in the number of Lamp1 positive structures (Supplementary Fig. 2a). Further, the depletion of endogenous Rab7 using siRNA led to a decrease in three-way contacts (Fig. 2c, d) suggesting that Rab7 is required for the formation of a subset of three-way contacts between the three organelles.

To test whether Rab7 was actively involved in the formation of the three-way contacts, we used the constitutively active mutant Rab7 Q67L that is locked at lysosomes and the dominant negative mutant Rab7 T22N that abolishes Rab7 localization at lysosomes (Fig. 2e). The overexpression of 3HA-Rab7 Q67L had no detectable effect on the proportion of three-way contacts compared to that of the wild-type 3HA-Rab7, but overexpression of the dominant negative 3HA-Rab7 T22N led to a significant reduction of lysosome-mitochondria contacts marked by the ER (Fig. 2f). Thus, our data suggest that lysosome localized Rab7 actively promotes the formation of the three-way contacts between lysosomes, mitochondria, and the ER.

**ORP1L mediates the Rab7 dependent ER-lysosome-mitochondria contact.** Oxysterol binding protein related protein 1L (ORP1L) is a lipid transfer protein that belongs to the family of Oxysterol-binding protein (OSBP)-related proteins (ORPs). ORP1L is recruited to lysosomes via its ankyrin repeat domains that bind to Rab7[18]. It also possesses an FFAT motif—two phenylalanines (FF) in an Acidic Tract—that allows it to interact with the ER resident VAPA and VAPB (hereafter referred to as VAPs) proteins to aid in establishing the ER-lysosomes contacts[12,19]. Interestingly, ORP1L was identified in the mitochondrial proteomic analysis MitoCarta3.0[20] and a rare genetic variant of *OSBPL1A* that encodes ORP1L was linked to mitochondrial dysfunctions[21], suggesting a possible role of ORP1L in mitochondrial function.

To test whether ORP1L has a role in the formation of the three-way contact during mitochondrial division, we first examined the subcellular localization of ORP1L positive lysosomes with respect to mitochondria and the ER. Consistent with previous findings we found that ORP1L localized to a subset of lysosomes marked by Lamp1 (Supplementary Fig. 2b) or Rab7 (Supplementary Fig. 2c). When co-expressed with the dominant negative Rab7 T22N, ORP1L remained cytosolic, confirming that its recruitment to lysosomes is Rab7 dependent (Supplementary

Fig. 2c). We tested whether ORP1L also localized on the ER-lysosome-mitochondrion three-way contact and found that GFP-ORP1L positive lysosomes appeared in contact with mitochondria (Fig. 3a) and localized to three-way contacts with the ER (Fig. 3b). We also found that contacts between ORP1L positive lysosomes and mitochondria (Fig. 3c) showed increased stability compared to Lamp1 positive lysosomes in general. This translated to an increased percentage and duration of ORP1L positive lysosomes contacts with mitochondria compared to Lamp1 positive lysosomes (Fig. 3d, e) suggesting that ORP1L overexpression could increase lysosome-mitochondria contacts frequency and duration.

Next, we asked whether the upregulation of the lysosome-mitochondria contact in cells overexpressing ORP1L was dependent on ORP1L interaction with the ER. We first compared the formation of lysosome-mitochondria contacts in cells overexpressing wild-type GFP-ORP1L or the FFAT motif mutant (D478A) that cannot act as an ER-lysosome tether[22] to cells overexpressing GFP. Here we monitored lysosome-mitochondria contacts, using Lamp1-mCherry as a lysosomal marker. The overexpression of ORP1L, but not that of the FFAT motif mutant, showed increase in both the proportion and duration of lysosome-mitochondria contacts (Fig. 3f, g) suggesting that ORP1L overexpression mediates lysosome-mitochondria contacts in an ER dependant manner.

Next, we asked whether ORP1L is required for the formation of ER-lysosome-mitochondria three-way contacts by disrupting the expression of ORP1L. In HeLa cells where ORP1L expression was either depleted (Supplementary Fig. 2d) or knocked out (Supplementary Fig. 2e), we observed a significant reduction but not an abolishment of three-way contacts (Fig. 3h, i). Importantly, ORP1L knockdown did not affect the proportion of lysosomes in contact with the ER (Supplementary Fig. 2f) or the percentage and duration of lysosome contacts with mitochondria (Supplementary Fig. 2g, h), indicating that the decreased proportion of ER-lysosome-mitochondria contacts was not caused by a global disruption of the ER-lysosome or lysosome-mitochondria contacts. Further, the siRNA studies also suggest that at basal conditions, ORP1L is not a dominant tethering protein for the lysosome-mitochondria contact. Next, we examined whether ORP1L's ER binding partners, VAPs, were required for the three-way contacts. Knockdown of VAPs resulted in a decrease in the three-way contacts (Fig. 3j and Supplementary Fig. 2i) suggesting that ORP1L interaction with VAPs plays an important role in the formation of the ORP1L dependent three-way contact.

To confirm that ORP1L was directly involved in the formation of a subset of three-way contacts, we artificially induced an acute

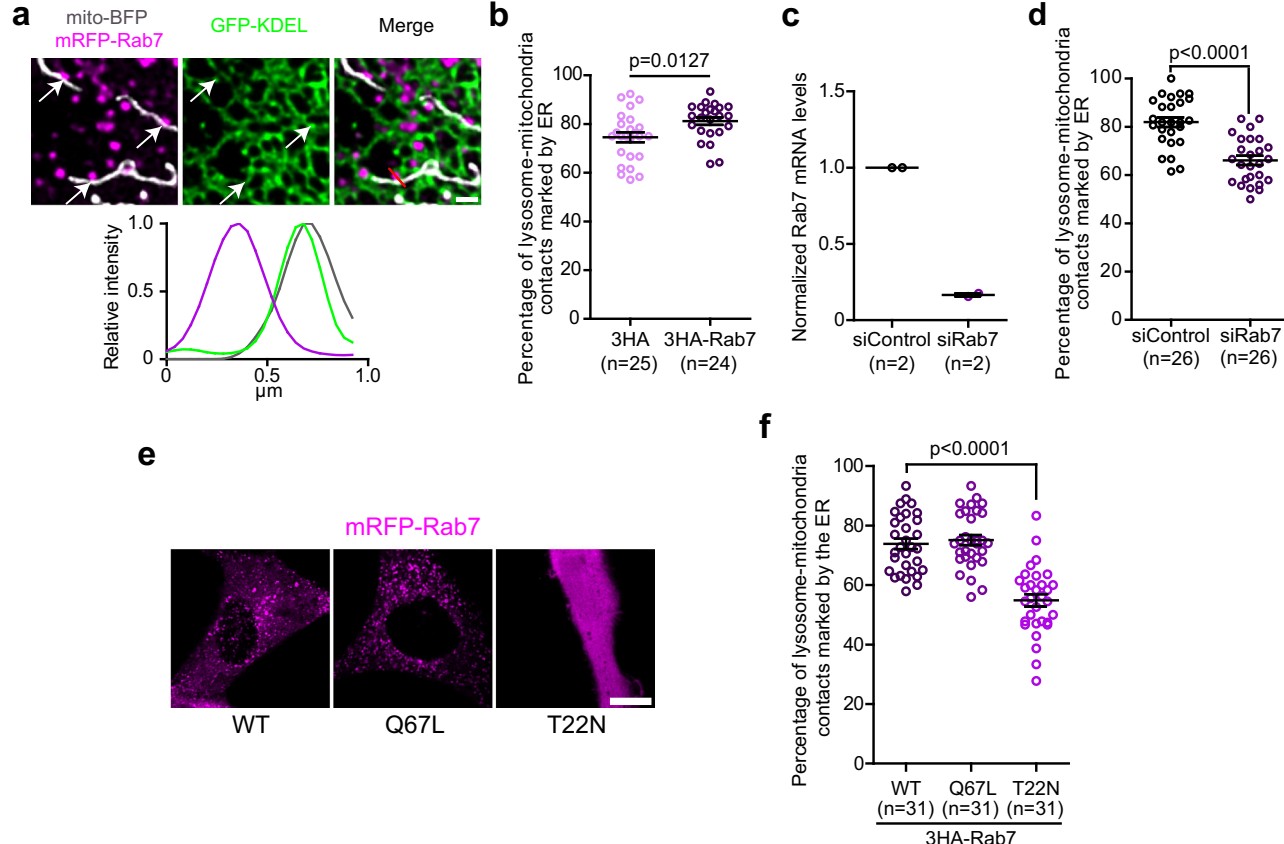

**Fig. 2 Rab7 regulates a subset of ER-lysosome-mitochondria three-way contacts. a** Representative image of Rab7 positive lysosomes-mitochondria contacts that are marked by the ER in HeLa cells expressing mRFP-Rab7, GFP-KDEL, and mito-BFP. White arrows indicate the lysosome-mitochondria contacts. Line-scan analysis of relative fluorescence intensities (left to right) from the red line shown in merge. Scale bar: 1 μm. **b** Quantification of the percentage of lysosome-mitochondria contacts marked by the ER in HeLa cells expressing either a wild-type Rab7 or a control vector, with mito-BFP, GFP-KDEL, and Lamp1-mCherry. The graphs show the mean ± SEM, cells from three independent experiments. Two-sided unpaired *t*-test. **c** Normalized Rab7 mRNA levels in cells transfected by a siRNA targeting specifically Rab7 or a control siRNA measured by RT-qPCR. The graphs show the mean ± SEM, cells from two independent experiments. **d** Quantification of the percentage of lysosome-mitochondria contacts marked by the ER when Rab7 expression level was downregulated by a siRNA targeting Rab7 compared to a control siRNA in cells expressing Lamp1-mCherry, mito-BFP, and GFP-KDEL. The graphs show the mean ± SEM, cells from three independent experiments. Two-sided unpaired *t*-test. **e** Representative images of HeLa cells expressing the wild-type (WT), constitutively active (Q67L), or dominant negative (T22N) mRFP-Rab7. Note that the dominant negative Rab7 (T22N) does not localize to lysosomes but is cytosolic. Scale bar: 10 μm. **f** Quantification of the percentage of lysosome-mitochondria contacts that are marked by the ER in HeLa cells expressing either the 3HA-Rab7 or its mutants with Lamp1-mCherry, mito-BFP, and GFP-KDEL. The graph shows the mean ± SEM, Cells from three independent experiments. One-way ANOVA with Dunnett's Multiple Comparison Test.

recruitment of ORP1L to Rab7 lysosomes in HeLa cells and monitored for ER at lysosome-mitochondria contacts as a way to evaluate the three-way contacts. We used the GID1-GAI dimerization system[23] where GAI-ΔANKORP1L, a GAI fused ORP1L construct lacking the ankyrin repeat domains and therefore cytosolic, was recruited to GID1-Rab7 positive lysosomes upon treatment with Gibberellin (GA₃-AM) that triggered the dimerization of GID1 with GAI (Fig. 3k, l). GAI-ΔANKORP1L D478A, the FFAT mutant version of ORP1L, was used as a negative control. A schematic summary of the methodology used for this experiment is shown in Supplementary Fig. 2j. Both GAI-ΔANKORP1L constructs were efficiently recruited to Rab7 lysosomes upon Gibberellin treatment (Fig. 3m), but the formation of three-way contacts was only promoted by the recruitment of the wild-type GAI-ΔANKORP1L (Fig. 3n).

Finally, to validate the specificity of the Rab7-ORP1L-VAPs axis in the formation of ER-lysosome-mitochondria contacts, we depleted another lysosomal lipid transport protein, STARD3, and monitored the three-way contacts. STARD3 also mediates ER-lysosome contacts via an interaction with VAPs, but

independently of Rab7[24], and was proposed to contribute to the formation of lysosome-mitochondria contacts[25]. We found that STARD3 knockdown did not recapitulate the loss of three-way contacts caused by the downregulation of ORP1L or VAPs (Supplementary Fig. 2k, l). Collectively, these data suggest that the Rab7-ORP1L-VAPs axis regulates the formation of a subset of three-way contacts between the ER, lysosomes, and mitochondria.

**The Rab7-ORP1L-VAPs axis regulates mitochondrial division.** We next tested whether the three-way contacts mediated by the Rab7-ORP1L-VAPs interaction was required for mitochondrial division. Recently, it has been shown that overexpression of constitutively active Rab7 Q67L impairs mitochondrial fission indicating that Rab7 GTP hydrolysis contributes to the division process[7]. Here, we found that both overexpressing either the GTP-locked Rab7 Q67L mutant and the GDP-locked Rab7 T22N dominant negative mutant reduced the rate of mitochondrial division (Fig. 4a). The decreased mitochondrial division rates, more particularly in cells overexpressing the T22N mutant, were

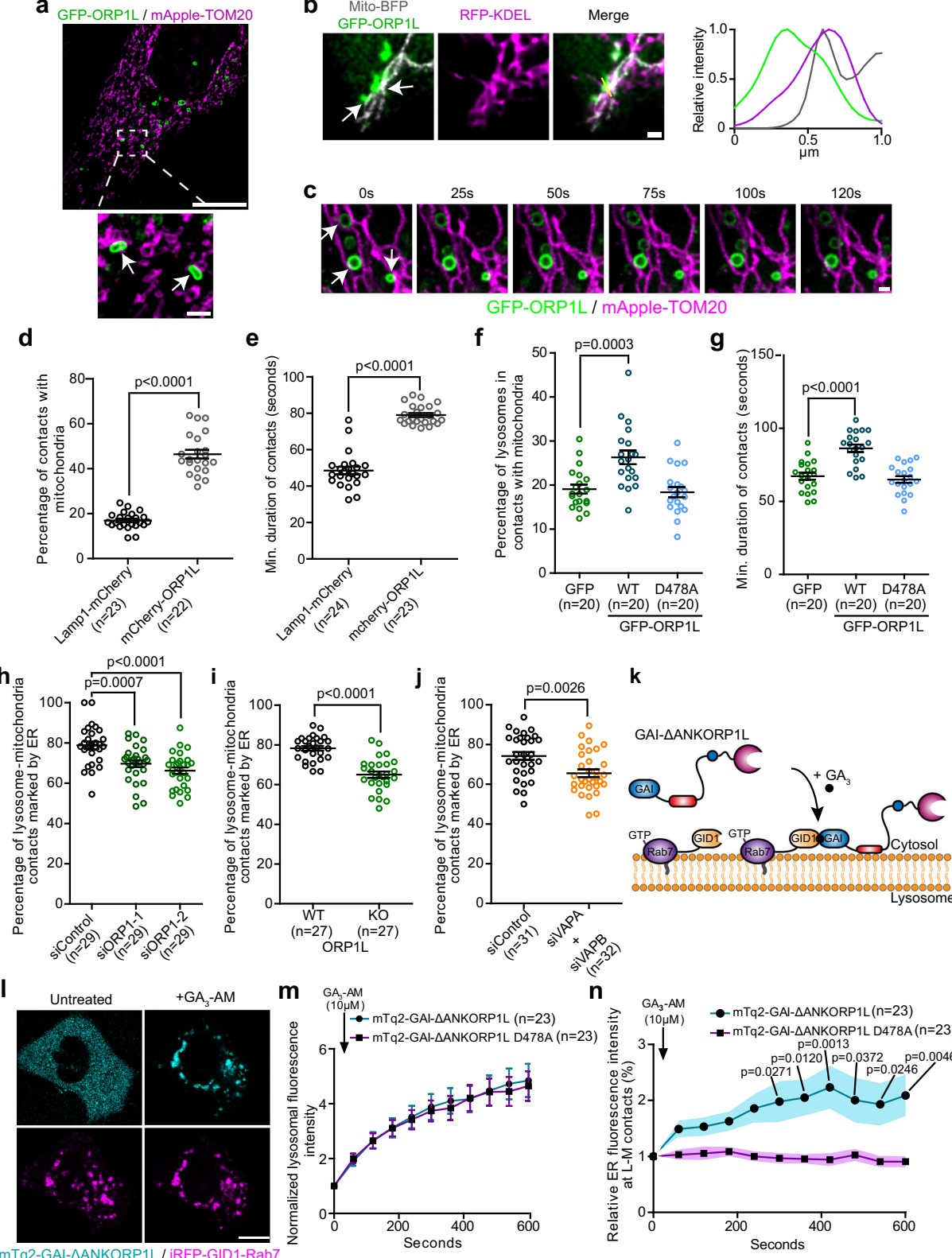

associated with an elongation of mitochondria, an increase of the number of mitochondrial junctions and a decrease in mitochondria numbers (Supplementary Fig. 3a–d) indicating a more hyperfused mitochondrial network. This suggests that not only is Rab7 GTP hydrolysis required for mitochondrial division, but that Rab7 must be localized at the lysosomes. A similar decrease in mitochondrial division rate was also observed in cells depleted

of VAPs (Fig. 4b). Together these results suggest that both Rab7 and VAPs are required for mitochondrial division.

To test whether ORP1L was also required for mitochondrial division, we first examined for ORP1L positive lysosomes at the site of fission. We found that ORP1L positive lysosomes were recruited to 58.1% ($n = 43/74$ events from 24 cells) of mitochondrial division events (Fig. 4c, d), which is similar to

**Fig. 3 The Rab7 interactor ORP1L regulates the formation of ER-lysosome-mitochondria three-way contacts in Hela cells. a** Representative image of a cell expressing GFP-ORP1L and mApple-TOM20 acquired by SIM microscopy. White arrows show ORP1L positive lysosomes-mitochondria contacts. Scale bars: 10 μm and 1 μm (inset). **b** Representative image of two ORP1L positive lysosome-mitochondria contacts, indicated by white arrows, marked by the ER. Line-scan analysis of relative fluorescence intensities from the yellow line in merge is shown. Scale bar: 1 μm. **c** Representative time lapse image of ORP1L positive lysosomes forming stable contact with mitochondria indicated by white arrows on the first frame. Scale bar: 1 μm. **d**, **e** Percentage of Lamp1-mCherry and mCherry-ORP1L positive lysosomes in contact with mitochondria (**d**) and **e** minimum duration of these contacts in cells co-expressing mito-GFP. Cells from three independent experiments. Two-sided unpaired *t*-test. **f**, **g** Percentage of Lamp1-mCherry positive lysosomes in contact with mitochondria (mito-BFP) (**f**) and **g** minimum duration of these contacts in cells expressing GFP, GFP-ORP1L, or the D478A mutant. Cells from two independent experiments. One-way ANOVA with Dunnett's Multiple Comparison Test. **h**–**j** Percentage of lysosome-mitochondria contacts that were marked by the ER: **h** when ORP1 expression was downregulated. Cells from three independent experiments, one-way ANOVA with Dunnett's Multiple Comparison Test; **i** in ORP1L KO cells and control cells. Cells from three independent experiments, two-sided unpaired *t*-test; and **j** when VAPA and VAPB expression levels were downregulated. Data from three independent experiments, Two-sided unpaired *t*-test. **k** Cytosolic GAI-ΔANKORP1L can be recruited to GID1-Rab7 lysosomes upon GA₃-AM treatment. **l** Representative image of a cell expressing mTq2-GAI-ΔANKORP1L and iRFP-GID1-Rab7 before and after GA₃-AM treatment (10 μM). Scale bar: 10 μm. **m** Normalized fluorescence intensity of mTq2-GAI-ΔANKORP1L WT and D478A colocalizing with iRFP-GID1-Rab7 lysosomes at the indicated times before and after GA₃-AM treatment. Cells from four independent experiments. **n** Relative ER fluorescence intensity at lysosome-mitochondria contacts at the indicated times before and after GA₃-AM treatment (shaded area represent the area within one SEM). Cells from four independent experiments. Two-way ANOVA, Sidak's multiple comparisons test. **d**–**j**, **m**, **n** All graphs show the mean ± SEM.

the proportion of division events marked by lysosomes (Fig. 1e). In fact, we found that almost 90% of lysosomes marking mitochondrial division possessed GFP-ORP1L (Supplementary Fig. 3e). When cells were depleted of ORP1L, a robust alteration of the mitochondrial morphology was observed with more elongated and hyperfused mitochondria (Fig. 4e–h) that was associated with a decreased rate of division and no difference in fusion rate compared to cells treated with a control siRNA (Fig. 4i and Supplementary Fig. 3f). A similar decrease in the division rate was also observed in ORP1L KO HeLa cells (Fig. 4j). These alterations were not caused by diminished levels of key components of the fission machinery such as Drp1, as its protein level in ORP1L KO cells was unchanged (Supplementary Fig. 3g). Similarly, siRNA depletion of ORP1L did not affect the mRNA levels of Drp1 or its adapter MFF (Supplementary Fig. 3h, i). Moreover, stimulated Drp1 dependent mitochondrial fission induced by either carbonyl cyanide 3-chlorophenylhydrazone (CCCP) treatment or by the overexpression of mitochondrial anchored protein ligase (MAPL)[26] was strongly reduced in cells depleted of ORP1L (Supplementary Fig. 4a–h). Collectively, these data strongly suggest that the Rab7-ORP1L-VAPs interaction is required for the efficient division of mitochondria.

**ORP1L Lipid transport domain is required for mitochondrial division.** A key functional domain of ORP1L is its lipid transport domain called the OSBP-related domain (ORD) that is required for the transfer of cholesterol[19] and PI(4)P[27]. To test whether ORP1L's lipid transfer activity is required for mitochondrial division or whether it simply acts as a tether for the ER-lysosome-mitochondrion three-way contact, we performed complementation experiments on the ORP1L KO HeLa cells by reintroducing either the wild-type ORP1L or functional domain mutants and quantified the rate of mitochondrial division. The three different ORP1L mutants were: (i) the ΔORD construct which lacks the lipid transfer domain, (ii) the HH/AA mutant that has two mutations in the ORD domain abolishing its ability to extract PI(4)P from membranes[27], and (iii) the FFAT motif D478A mutant that is unable to interact with VAPs[27] (Fig. 4k and Supplementary Fig. 5a). Importantly, these constructs still localize to Rab7 positive lysosomes[27]. The later mutant served as a negative control as it cannot interact with the ER thus impairing formation of ER-lysosome-mitochondria contacts. Both the ΔORD and the HH/AA constructs were readily recruited to lysosome-mitochondria contact sites at a similar level as the wild-type (Supplementary Fig. 5b, c). The D478A mutant showed a

significant decrease in lysosome-mitochondria contact supporting the need of VAPs in the formation of the lysosome-mitochondria contact site (Supplementary Fig. 5c). However, only the wild-type ORP1L was able to improve the rate of mitochondrial division in ORP1L KO cells (Fig. 4l) suggesting that ORP1L roles in both the formation of three-way contacts and in the transfer of lipids are important for mitochondrial division.

To test whether the lipid transfer function of ORP1L was directly involved in the fission and rule out the possibility of an indirect effect of a loss in ORP1L function, we acutely recruited ORP1L to lysosomes in ORP1L KO cells using the GID1-GAI system and monitored the mitochondrial division rate. The recruitment of GAI-ΔANKORP1L to lysosomes increased the rate of mitochondrial division in ORP1L KO cells (Fig. 4m) suggesting that an acute rescue of the ORP1L mediated lipid transfer was sufficient to restore mitochondrial division. Together, our rescue experiments show that ORP1L is required not only to form an ER-lysosome-mitochondrion three-way contact at the site of mitochondrial fission, but also that its lipid transfer domain is required for mitochondrial division suggesting that ORP1L transfer lipids during the mitochondrial fission process.

**Lysosomal PI(4)P is required for mitochondrial division.** ORP1L was previously shown to mediate the transfer of cholesterol from lysosomes to the ER[19] and PI(4)P from phagolysosomes to the ER[27]. Interestingly, among lysosomes marking the mitochondrial division site, 83.8% ($n = 26/31$ events from 22 cells; Supplementary Fig. 5d, e) were positive for the cholesterol probe D4H[28] and 82.1% ($n = 32/39$ events from 21 cells; Fig. 5a, b) were positive for the PI(4)P probe 2xP4M[29]. To test whether cholesterol transfer was required in the division of mitochondria we depleted NPC1—a key protein in the egress of cholesterol from lysosomes[25]—using siRNA (Supplementary Fig. 5f), as ORP1L was shown to require NPC1 to transport lysosomal cholesterol to the ER[19]. We found that NPC1 down-regulation had no effect on mitochondrial morphology or on the mitochondrial division rate (Supplementary Fig. 5g–k), indicating that ORP1L was not required to transfer cholesterol from the lysosomes to the site of mitochondrial division.

As PI(4)P production at the Golgi was recently shown to play a crucial role in mitochondrial division[6] and lysosomes marking mitochondrial division events were positive for the PI(4)P probe 2xP4M (Fig. 5a, b), we hypothesized that ORP1L may mediate the transfer of PI(4)P from lysosomes to the site of mitochondrial

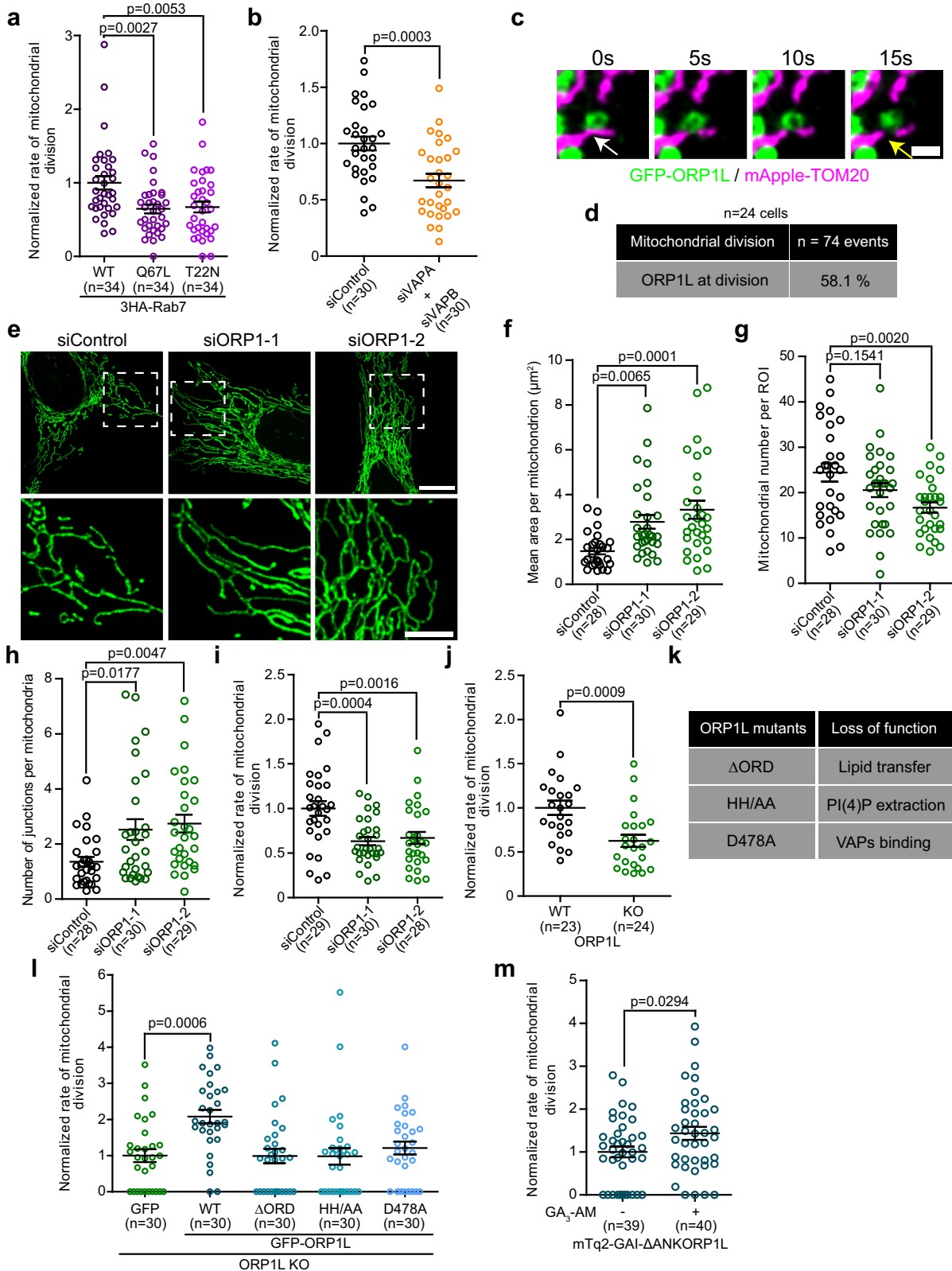

division. To test this, we used a previously validated chimeric construct called ORPSAC1[27], where the ORD domain of ORP1L was replaced by the catalytic domain of Sac1, a phosphatase that dephosphorylates PI(4)P. This allowed us to test whether the specific depletion of PI(4)P at the cytosolic leaflet of the lysosomal membrane impairs mitochondrial division (Fig. 5c). The overexpression of both wild-type ORP1L and ORPSAC1 led to a

significant reduction of the 2xP4M signal at ORP1L positive lysosomes compared to the overexpression of the HH/AA mutant ORP1L that is unable to extract PI(4)P (Supplementary Fig. 6a, b). These results confirm that ORPSAC1 is able to deplete PI(4)P at lysosomes and that, as was previously reported[27], ORP1L regulates the levels of PI(4)P at lysosomes, likely by transferring it to other compartments. Next, we examined the morphology of

**Fig. 4 The Rab7-ORP1L-VAPs interaction regulates mitochondrial division. a, b** Normalized rate of mitochondrial division (number of mitochondrial divisions normalized by time and volume) in HeLa cells (**a**) expressing wild-type or mutants 3HA-Rab7. Cells from three independent experiments. One-way ANOVA with Dunnett's Multiple Comparison Test; **b** when VAPs expression levels were downregulated using siRNAs. Cells from three independent experiments. Two-sided unpaired *t*-test. The graphs show the mean ± SEM. **c** Live-cell imaging of HeLa cells expressing GFP-ORP1L and mApple-TOM20. The white and yellow arrows show an ORP1L positive lysosome-mitochondria contact and a mitochondrial fission event. Scale bar: 1 μm. **d** Percentage of mitochondrial division events that were marked by GFP-ORP1L (cells from two independent experiments) in HeLa cells. **e** Representative maximum projection images of mitochondrial morphology in HeLa cells treated with indicated siRNAs. Scale bars: 10 μm and 5 μm (inset). **f–h** Mitochondrial morphology was quantified for **f** mean area per mitochondrion, **g** mitochondrial number per region of interest (ROI), and **h** number of junctions per mitochondria. Cells from three independent experiments. One-way ANOVA with Dunnett's Multiple Comparison Test. The graphs show the mean ± SEM. **i, j** Normalized rate of mitochondrial division of **i** HeLa cells treated with indicated siRNAs. Cells from three independent experiments. One-way ANOVA with Dunnett's Multiple Comparison Test and of **j** ORP1L WT and KO HeLa cells. Cells from three independent experiments. Two-sided unpaired *t*-test. The graphs show the mean ± SEM. **k** ORP1L mutants used and the loss of function caused by the mutations or deletions. **l** Normalized rate of mitochondrial division in ORP1L KO HeLa cells expressing GFP or GFP-ORP1L constructs and mApple-TOM20. The graphs show the mean ± SEM, cells from three independent experiments. One-way ANOVA with Dunnett's Multiple Comparison Test. **m** Normalized rate of mitochondrial division in ORP1L KO HeLa cells expressing mTq2-GAI-ΔANKORP1L, iRFP-GID1-Rab7, and mApple-TOM20 before and after GA₃-AM (10 μM) treatment. Cells were imaged from 5 to 20 min after GA₃-AM treatment. Cells from three independent experiments. Two-sided unpaired *t*-test. The graphs show the mean ± SEM.

mitochondria in cells overexpressing GFP-ORPSAC1 and found that it was profoundly elongated and hyperfused (Fig. 5d–g), and these cells showed a significantly reduced rate of mitochondrial division (Fig. 5h) compared to cells overexpressing GFP or GFP-ORP1L. ORPSAC1 positive lysosomes were in contact with mitochondria for the same frequency and duration as wild-type ORP1L positive lysosomes suggesting that its overexpression does not alter lysosome-mitochondria contacts (Supplementary Fig. 6c, d), nor did it impair the localization of ORP1L at lysosomes (Supplementary Fig. 6e). Additionally, ORPSAC1 positive lysosomes were observed in contact with mApple-Drp1 puncta at mitochondrial constriction sites (Supplementary Fig. 6f) indicating that the depletion of PI(4)P was not preventing the recruitment of lysosomes to mitochondrial constriction sites.

To confirm the need for lysosomal PI(4)P in mitochondrial fission and rule out a potential effect of a long-term depletion of lysosomal PI(4)P that may indirectly affect mitochondrial division, we acutely depleted PI(4)P at Rab7 positive lysosomes by targeting a cytosolic Sac1 to lysosomes using the GID1-GAI system (Fig. 5i). Recruitment of wild-type Sac1, but not of the catalytic dead C392S mutant, led to a decrease of the 2xP4M levels at lysosomes (Fig. 5j, k) as well as a decrease in the rate of mitochondrial division (Fig. 5l) demonstrating that an acute depletion of PI(4)P at the lysosomal membrane impairs mitochondrial division. Finally, we tested whether inhibiting PI(4)P production at the lysosome subdued mitochondrial division. Lysosomal PI(4)P was proposed to be generated by type II phosphatidylinositol 4-kinases (PI4K) such as PI4K2A and PI4K2B[30–32]. GFP tagged PI4K2B localized to lysosomes (Fig. 5m) and downregulation of PI4K2B (Supplementary Fig. 7a) led to decreased levels of 2xP4M at lysosomes (Supplementary Fig. 7b) consistent with a role of PI4K2B in the production of lysosomal PI(4)P. Quantification of the mitochondrial division rate in the PI4K2B knockdown cells showed a depressed division rate and a significantly hyperfused mitochondrial morphology (Fig. 5n and Supplementary Fig. 7c–f) compared to cells treated with a control siRNA. Collectively, our results indicate a crucial role for lysosomal PI(4)P in mitochondrial fission and suggest that ORP1L mediates PI(4)P transfer during this process.

**Restoring PI(4)P transfer from lysosomes rescues mitochondrial morphology in ORP1L depleted cells.** To confirm that a transfer of lysosomal PI(4)P across the ER-lysosome-mitochondrion three-way contact was required for mitochondrial division, we depleted HeLa cells of ORP1L using siRNA and tested whether re-establishing the PI(4)P transfer from lysosomes was able to rescue the morphology of mitochondria caused by

ORP1L depletion. To do so, we generated a chimeric protein that we called ORPOSBP in which the N-terminus of OSBP was replaced by the N-terminus of ORP1L that contains the Rab7 binding Ankyrin repeats (Fig. 6a). OSBP is a well-established PI(4)P transfer protein that mediates its transfer at the trans-Golgi network–ER[33,34] and at endo-lysosomes-ER contact sites[35,36].

When overexpressed in cells, ORPOSBP almost exclusively localized to Rab7 positive lysosomes (Fig. 6b) that were found at mitochondrial fission sites (Fig. 6c). Further, cells expressing ORPOSBP showed lower 2xP4M levels at lysosomes compared to a HH/AA mutant that abolishes OSBP ability to extract PI(4)P from membranes[37] (Fig. 6d, e), demonstrating that ORPOSBP is able to promote the transfer of PI(4)P from lysosomes. Importantly, expression of ORPOSBP, but not the HH/AA mutant, was sufficient to rescue the alterations of the mitochondrial morphology induced by the RNAi mediated depletion of ORP1L (Fig. 6f–i). However, ORPOSBP had no effect on mitochondrial morphology in control cells (Fig. 6j–l), similar to overexpression of ORP1L (Fig. 5e–g). Thus, our data is supportive of ORP1L mediating a PI(4)P transfer from lysosomes to promote mitochondrial fission.

**The recruitment of the PI(4)P phosphatase Sac1 to mitochondria impairs their division.** We next examined whether the mitochondrial division site may be the recipient of the lysosomal PI(4)P during mitochondrial division. Since any PI(4)P transferred to the ER is quickly converted into PI by the ER localized SAC1[38], we reasoned that ORP1L likely transfers PI(4)P to the mitochondrial membrane during mitochondrial division. To test this hypothesis, we first asked whether PI(4)P was required for mitochondrial fission by depleting PI(4)P at the division site by targeting of the catalytic domain of Sac1 using the GAI-GID1 dimerization system, where GID1 was fused to the mitochondrial outer membrane protein MFF that recruits Drp1 to the fission site[9] (Fig. 7a, b). The acute recruitment of the Sac1 catalytic domain, but not of the catalytic dead mutant C392S, to mitochondrial division sites led to a marked decrease of the mitochondrial division rate (Fig. 7c). Moreover, 6 h after targeting Sac1 to mitochondrial division sites resulted in a significant increase in mitochondria elongation and the hyperfusion of the mitochondrial network (Supplementary Fig. 8a–d). However, the recruitment of the catalytically dead Sac1 C392S did not affect mitochondrial morphology, indicating that the observed effects were dependent on Sac1 catalytic activity. To further validate the need for PI(4)P at the site of division, we expressed Sac1-MFF fusion protein in order to anchor Sac1 catalytic domain at the

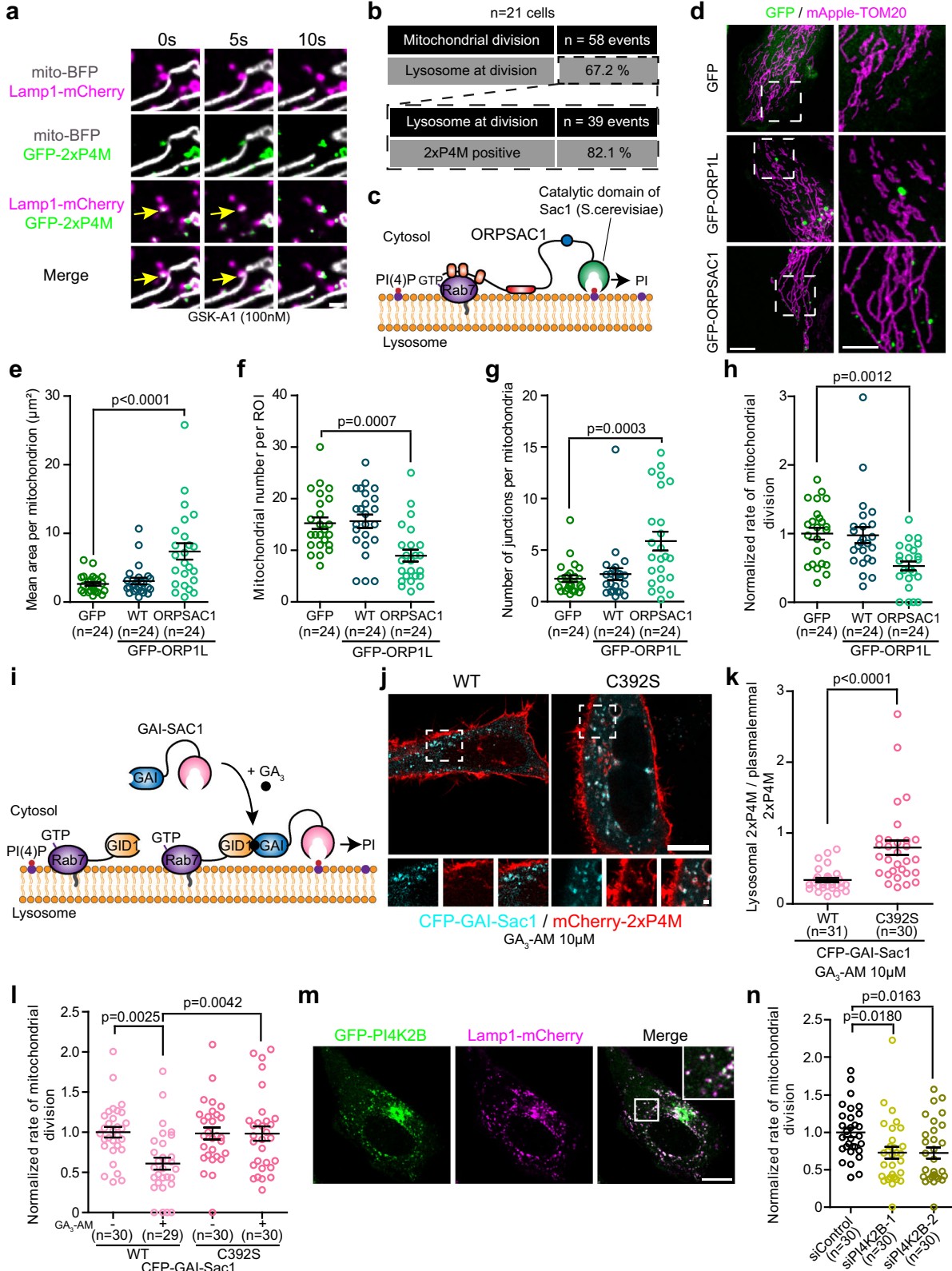

mitochondrial outer membrane (Fig. 7d). The GFP-Sac1-MFF expressing cells showed increased elongation and hyperfusion of the mitochondrial network (Fig. 7e–h) compared to control cells overexpressing GFP alone, the catalytic dead GFP-Sac1 C392S-MFF or GFP-MFF. Together, these results suggest that PI(4)P is required at the mitochondrial fission site and plays a crucial role in the division process.

To test whether production of PI(4)P at the mitochondrial division site was sufficient to promote fission, we targeted the catalytic domain of PI4KA (PI4KAc1001) to the division site (Supplementary Fig. 8e) and monitored the division rate in both control and in ORP1L KO cells. As previously demonstrated[39], the recruitment of PI4KA (PI4KAc1001) catalytic domain to mitochondria leads to PI(4)P production at the mitochondria

**Fig. 5 Lysosomal PI(4)P contributes to mitochondrial division. a** Representative image of a mitochondrial division marked by a GFP-2xP4M positive lysosome (yellow arrow) in HeLa cells expressing the indicated markers. PI4KA inhibitor GSK-A1 treatment allowed to better visualize PI(4)P at lysosomes. Scale bar: 1 μm. **b** Percentage of mitochondrial division events marked by lysosomes (cells from two independent experiments) and of lysosomes marked by GFP-2xP4M at mitochondrial division events. **c** In ORPSAC1, the ORD domain of ORP1L was replaced by the catalytic domain of Sac1 allowing the dephosphorylation of lysosomal PI(4)P. **d** Representative maximum projection images of mitochondrial morphology in HeLa cells overexpressing the indicated constructs. Scale bars: 10 μm and 5 μm (inset). **e–g** Mitochondrial morphology was quantified for **e** mean area per mitochondrion, **f** mitochondrial number per region of interest (ROI), and **g** number of junctions per mitochondria. Cells from three independent experiments. One-way ANOVA with Dunnett's Multiple Comparison Test. **h** Normalized rate of mitochondrial division in HeLa cells overexpressing the indicated constructs. Cells from three independent experiments. One-way ANOVA with Dunnett's Multiple Comparison Test. **i** Cytosolic GAI-Sac1 can be recruited to GID1-Rab7 lysosomes upon $GA_3$-AM treatment to specifically deplete lysosomal PI(4)P. **j, k** Representative images of HeLa cells expressing CFP-GAI-Sac1 or the catalytic dead C392S mutant, iRFP-GID1-Rab7, and mCherry-2xP4M after $GA_3$-AM treatment (**j**). Scale bars: 10 μm and 1 μm (inset). **k** Quantification of the lysosomal levels, normalized by the plasmalemmal levels, of 2xP4M. Cells from three independent experiments. Two-sided unpaired *t*-test. **l** Normalized rate of mitochondrial division before or after $GA_3$-AM treatment in HeLa cells overexpressing iRFP-GID1-Rab7 and the indicated constructs. Cells from three independent experiments. Two-way ANOVA, Sidak's multiple comparisons test. **m** Representative image of a HeLa cell expressing GFP-PI4K2B and Lamp1-mCherry with zoomed insert of white box in merge panel. Scale bar: 10 μm. **n** Normalized rate of mitochondrial division in HeLa cells treated with the indicated siRNAs. Cells from three independent experiments. One-way ANOVA with Dunnett's Multiple Comparison Test. **e–h, k, l, n** All graphs show the mean ± SEM.

(Supplementary Fig. 8f, g). However, it did not affect the mitochondrial division rate in control or in ORP1L KO cells (Supplementary Fig. 8h, i), suggesting that an increase in PI(4)P alone on mitochondria is not sufficient to promote their division. This result suggest that the timed increase of PI(4)P at the mitochondrial division site may be critical for orchestrating mitochondrial division.

**VAPs independent ORP1L lipid transport is sufficient to rescue mitochondrial division in ORP1L KO cells.** ORP1L lipid transfer between the ER and lysosome at the Rab7-ORP1L-VAPs mediated ER-lysosome contact site requires ORP1L binding to VAPs[19]. However, Rab7 is also involved in forming membrane contact site with mitochondria as its interaction with the mitochondrial localized protein TBC1D15 drive its GTP hydrolysis to untether the two organelles[7]. Since ORP1L binds directly to Rab7, we propose that ORP1L may directly transport lysosomal PI(4)P to mitochondria at the lysosome-mitochondria contacts induced by the recruitment of lysosomes to the site of mitochondrial division. To test this hypothesis, we examined whether a lysosomal localized ORP1L that cannot bind to VAPs, but still able to bind to Rab7, can rescue mitochondrial fission in the ORP1L KO cells. Since ORP1L-VAPs interaction is required for lysosome recruitment to the site of mitochondrial division, we co-transfected the ORP1L KO cells with both the FFAT mutant ORP1L D478A and the ORD deletion mutant ORP1LΔORD. Both constructs cannot rescue ORP1L KO cells individually (Fig. 4l). To distinguish the activity of these constructs we targeted the FFAT mutant (GAI-ΔANKORP1L D478A) to lysosomes using the GID1-GAI dimerization system. We found that only cells with both constructs at lysosomes showed increased mitochondrial division rate in the ORP1L KO cells (Fig. 7i, j). However, GAI-ΔANKORP1L D478A alone at the lysosome was not able to rescue mitochondria fission (Fig. 7k). These results suggest that ORP1L can transfer lipids without binding to VAPs thus giving further support for ORP1L as a PI(4)P transporter at the lysosome-mitochondrion contact during mitochondrial division.

## Discussion

Late endosomes/lysosomes (referred here as lysosomes) are recruited to the mitochondrial fission site but their role in the highly coordinated steps of mitochondrial division is unknown. Using super-resolution live-cell imaging we systematically analyzed the lysosomes at the mitochondrial division site, which

resulted in a number of new insights into their role during mitochondrial division (Fig. 8). First, we show that the ER recruit lysosomes to the site of mitochondrial division to form a three-way contact between the three organelles after Drp1 recruitment. Second, the formation of this three-way contact at the division site is mediated by the Rab7-ORP1L-VAPs interaction that bring lysosomes to the mitochondrial division site. Lastly, our work suggests that ORP1L is required for the transfer of PI(4)P from lysosomes to the site of mitochondrial division. Inhibiting this transfer or depleting PI(4)P at the division site impairs mitochondrial fission, indicating a need for PI(4)P in mitochondrial division.

Here, we show that lysosomes are recruited to a late stage of the mitochondrial division process by the formation of an ER-lysosome-mitochondrion three-way contact to aid in the fission of mitochondria. We propose that this is mediated by the formation of a Rab7-ORP1L-VAPs ER-lysosome contact in close proximity to the mitochondrial constriction site. Given that ORP1L binding to VAPs is required for this three-way contact, we propose that the ER acts as a regulator of lysosome-mitochondria contacts during mitochondrial division by recruiting lysosomes after the recruitment of Drp1. However, it should be noted that not all three-way contacts participate in mitochondrial division. Indeed, it was recently proposed that the ER resident protein PDZD8 mediates the formation of an ER-lysosome-mitochondrion three-way contact made of two contact sites (ER-lysosomes and ER-mitochondria) that serves to link the three organelles[15]. However, the PDZD8 mediated contact sites does not appear to have a role in mitochondrial division but instead seems to be required for $Ca^{2+}$ exchange between the ER and mitochondria[40] and to participate in endo-lysosome functions[15,41]. The existence of the PDZD8 mediated three-way contact may explain why the depletion of ORP1L or VAPs reduced but did not abolish ER-lysosome-mitochondria contacts. Therefore, we suggest that the Rab7-ORP1L-VAPs interaction mediates the formation of a subset of three-way contacts. Although it is possible that the Rab7-ORP1L-VAPs mediated three-way contacts may be involved in other functions, it appears to preferentially contribute to mitochondrial division. We find that the majority of lysosomes at the site of mitochondrial fission are positive for ORP1L, and disruption of the Rab7-ORP1L-VAPs interaction not only results in a decrease of the three-way contacts but also strongly impairs mitochondrial division. However, consistent with previous studies showing lysosomes at a majority but not all division events[6,7], the Rab7-ORP1L-VAPs mediated three-way contact is not absolutely essential for mitochondrial

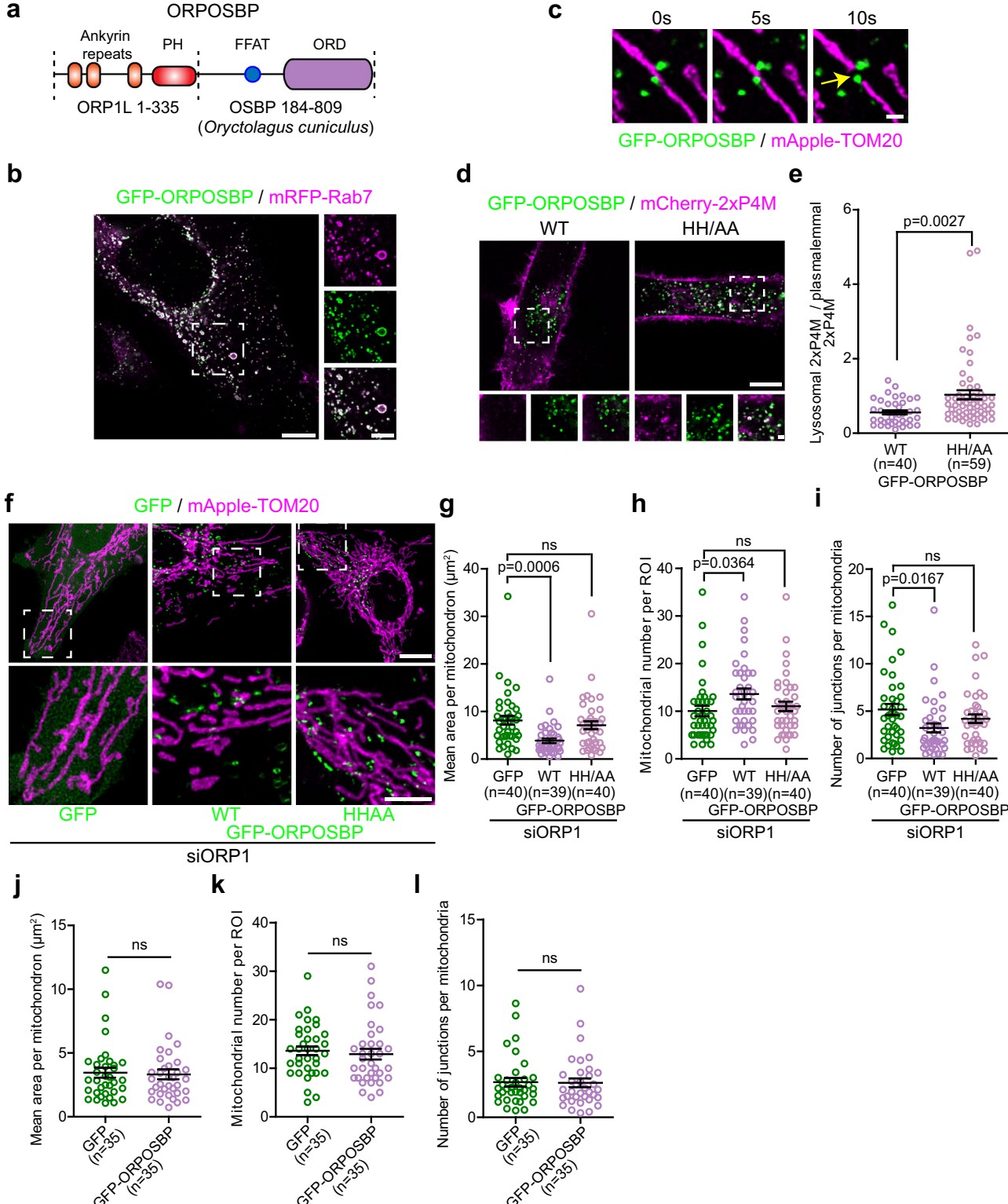

fission as disrupting their formation did not abolish mitochondrial division but reduced the division rate. Indeed, we report a decrease of almost 50% of the mitochondrial division rate when the Rab7-ORP1L-VAPs interaction is impaired which is similar and consistent with the percentage of mitochondrial division showing lysosomes (and thus ER-lysosome-mitochondria contact) recruitment.

The role of ORP1L in mitochondrial division appears to be more than tethering lysosomes to the site of mitochondrial

division. ORP1L mutants that lack the lipid transfer domain or that cannot extract PI(4)P from membranes did not rescue mitochondrial division in ORP1L KO cells. This suggests that ORP1L regulates PI(4)P at the ER-lysosome-mitochondrion three-way contact during the mitochondrial division process. ORP1L may be regulating PI(4)P pools on the lysosome, the ER, or mitochondria during mitochondrial division. While we did not unequivocally demonstrate that ORP1L transfer PI(4)P at the three-way contact, our results suggest a model where ORP1L

**Fig. 6 Targeting the PI(4)P transfer protein OSBP to lysosomes rescues the mitochondrial morphology alterations caused by ORP1L depletion. a** ORPOSBP is a chimeric protein made of the amino acids 1 to 335 of ORP1L fused to amino acids 184 to 809 of OSBP from *Oryctalogus cuniculus*. Domains are not to scale. **b** Representative image of a Hela cell expressing GFP-ORPOSBP and mRFP-Rab7. Zoomed images of dashed box shown on right. Scale bars: 10 μm and 5 μm (inset). **c** Time-lapse images of HeLa cells expressing GFP-ORPOSBP and mApple-TOM20. The yellow arrow shows a mitochondrial fission event marked by GFP-ORPOSBP. Scale bar: 1 μm. **d**, **e** Representative images of HeLa cells expressing the wild-type GFP-ORPOSBP or the lipid binding deficient HH/AA mutant with the PI(4)P probe mCherry-2xP4M (**d**). Scale bars: 10 μm and 1 μm (inset). **e** Quantification of the lysosomal levels of 2xP4M normalized by the plasmalemmal levels of the probe. The graphs show the mean ± SEM, cells from three independent experiments. Two-sided unpaired *t*-test. **f** Representative maximum projection images of mitochondrial morphology in HeLa cells treated with siRNA downregulating ORP1L and overexpressing GFP, GFP-ORPOSBP or the HH/AA ORPOSBP mutant and mApple-TOM20. Note that ORPOSBP wild-type expression, but not of the HH/AA mutant rescue the mitochondrial elongation and hyperfusion of the mitochondrial network caused by ORP1L depletion. Scale bars: 10 μm and 5 μm (inset). **g–i** Mitochondrial morphology was quantified for **g** mean area per mitochondrion, **h** mitochondrial number per region of interest (ROI), and **i** number of junctions per mitochondria. Cells from three independent experiments. One-way ANOVA with Dunnett's Multiple Comparison Test, ns non statistically significant: **g** $p = 0.6437$, **h** $p = 0.7205$, and **i** $p = 0.3046$. The graphs show the mean ± SEM. **j–l** Mitochondrial morphology of Hela cells expressing GFP or GFP-ORPOSBP was quantified for **j** mean area per mitochondrion, **k** mitochondrial number per region of interest (ROI), and **l** number of junctions per mitochondria. Cells from three independent experiments. Two-sided unpaired *t*-test, ns non statistically significant. **j** $p = 0.8011$, **k** $p = 0.6381$ and **l** $p = 0.9266$. The graphs show the mean ± SEM.

provides PI(4)P to the mitochondrial division site by mediating its transfer from the lysosome. First we find that depleting lysosomal PI(4)P, or inhibiting PI(4)P production at lysosomes, thus preventing ORP1L mediated PI(4)P transfer, impairs mitochondrial division. Second, targeting OSBP, a bona fide PI(4)P transferring protein[33–36], to Rab7 positive lysosomes rescued mitochondrial morphology caused by ORP1L downregulation. Finally, targeting the PI(4)P phosphatase Sac1 to the mitochondrial division site impaired their division. Together, these results support a role for an ORP1L mediated PI(4)P transfer from lysosomes during mitochondrial division where the lysosomes provide PI(4)P to the site of fission at the final stages of mitochondrial division.

Recently, PI(4)P production at the Golgi apparatus and recruitment of Golgi-derived vesicles to the fission site were shown to be required for efficient mitochondrial division[6]. Although it has been speculated that these Golgi vesicles may provide PI(4)P to the division site[42], it has not been explicitly demonstrated. However, when we depleted PI(4)P either on lysosomes or on mitochondria, we observed a similar decrease in the mitochondrial division rate, suggesting that Golgi vesicles were not able to compensate for the loss of lysosomal PI(4)P at the lysosome-mitochondria division contact site. Since we find that lysosomes are recruited before Golgi vesicles to the division site, one possibility could be that the two organelles provide PI(4)P for different purposes or at different step of the fission event. Additionally, as different types of mitochondrial division were recently reported[43], lysosomes and Golgi vesicles may contribute to functionally distinct mitochondrial fissions. Further studies are required to understand the precise role of these two organelles and their interaction during the division process.

Finally, while PI(4)P production at the Golgi was recently showed to be crucial in mitochondrial division[6], our data suggest that PI(4)P may play a direct role in mitochondrial division. Indeed, targeting the PI(4)P phosphatase Sac1 to mitochondria impairs their division. Therefore, we propose a model where ORP1L mediates a PI(4)P transfer from lysosomes to the mitochondrial outer membrane at the division site to generates a transient PI(4)P signal on mitochondria that helps to execute membrane fission. While the role of this PI(4)P signal remains elusive, the previously described roles of PI(4)P in cells[44,45] suggest that PI(4)P could potentially participate in actin nucleation, recruitment or modulation of effectors, or affect mitochondrial membrane curvature at the division site to help the completion of mitochondrial membranes scission. Similarly, given that some ORPs transport lipids through lipid exchange[33,46] it is possible that mitochondrial division does not

only need PI(4)P but also requires the removal of another lipid at the fission site. As ORP1L's role in cholesterol transfer was previously reported[19], cholesterol levels may be modulated at the division site via an exchange with PI(4)P. The lipid exchange mechanism at the division site is supported in part by the PI(4)P/cholesterol exchanger OSBP rescuing mitochondrial morphology upon being targeted to lysosomes in ORP1L depleted cells. We also find that the timing of PI(4)P at the site of division is likely critical for orchestrating mitochondrial division events as the synthesis of PI(4)P on mitochondria did not promote their division. Thus, giving further support for the timed recruitment of lysosomes to the site of division. Further studies are required to decipher the precise role of PI(4)P in the mitochondrial division process.

In conclusion, we show that during mitochondrial division, the ER recruits lysosomes to the site of mitochondrial division via the Rab7-ORP1L-VAPs tethering complex resulting in the formation of a three-way contact between the three organelles. Here, the lipid transfer protein ORP1L may mediates a PI(4)P transfer from lysosomes to the division site in a mechanism independent of its VAPs binding motif. Inhibition of this transfer or depletion of PI(4)P at the mitochondrial fission site impairs mitochondrial division and alters their morphology. Our data, therefore, support that PI(4)P directly plays a vital role in mitochondrial division.

## Methods

**Antibodies and chemicals**. The following antibodies were used in this study: mouse anti-Lamp1 (DHSB, H4A3, dilution: 1:1000), rabbit anti-ORP1 (Abcam, ab131165, dilution: 1:1000), mouse anti-Drp1 (BD Biosciences, 611112, dilution: 1:1000), rabbit anti-VAPA (Novus Biologicals, NBP1-31237, dilution: 1:1000), rabbit-anti-VAPB (MilliporeSigma, HPA013144, dilution: 1:1000), mouse-anti GAPDH HRP conjugated (Novus Biologicals, NB300-328H, dilution: 1:10000), goat anti-Rabbit HRP (ThermoFisher, 31460, dilution: 1:10000), goat anti-Mouse HRP (Cedarlane, CLCC30007, dilution: 1:10000) and goat anti-Mouse Alexa 568 (ThermoFisher, A-11011, dilution: 1:1000). GA₃-AM (SML1959), Carbonyl cyanide 3-chlorophenylhydrazone (CCCP) (C2759), and GSK-A1 (SML2453) were purchased from MilliporeSigma. MitoTracker™ Deep Red FM - Special Packaging (M22426) was purchased from ThermoFisher.

**Plasmids**. The following plasmids were used in this study: GFP-KDEL, RFP-KDEL, mito-GFP and GFP-Drp1 were described previously[47,48]. mito-BFP was a gift from William Trimble (Hospital for Sick Children, Toronto); mApple-TOM20 was a gift from Michael Davidson (Addgene #54955); Lamp1-mCherry was a gift from Amy Palmer (Addgene #45147); Arf1-GFP was a gift from Paul Melancon (Addgene #39514); mRFP-Rab7, mRFP-Rab7 Q67L and mRFP-T22N were gifts from John Brumell (Hospital for Sick Children, Toronto); mCherry-2xP4M, GFP-2xP4M, GFP-ORP1L, mCherry-ORP1L, GFP-ORP1L ΔORD, GFP-ORP1L HH/AA, GFP-ORP1L D478A, GFP-ORPSAC1, mTq2-GAI-ΔANKORP1L, mTq2-GAI-ΔANKORP1L D478A, iRFP-GID1-Rab7, Lamp1-mEmerald and mCherry-D4H were gifts from Sergio Grinstein (Hospital for Sick Children, Toronto). The following constructs were made for this study: 3HA-Rab7, 3HA-Rab7, 3HA-Rab7

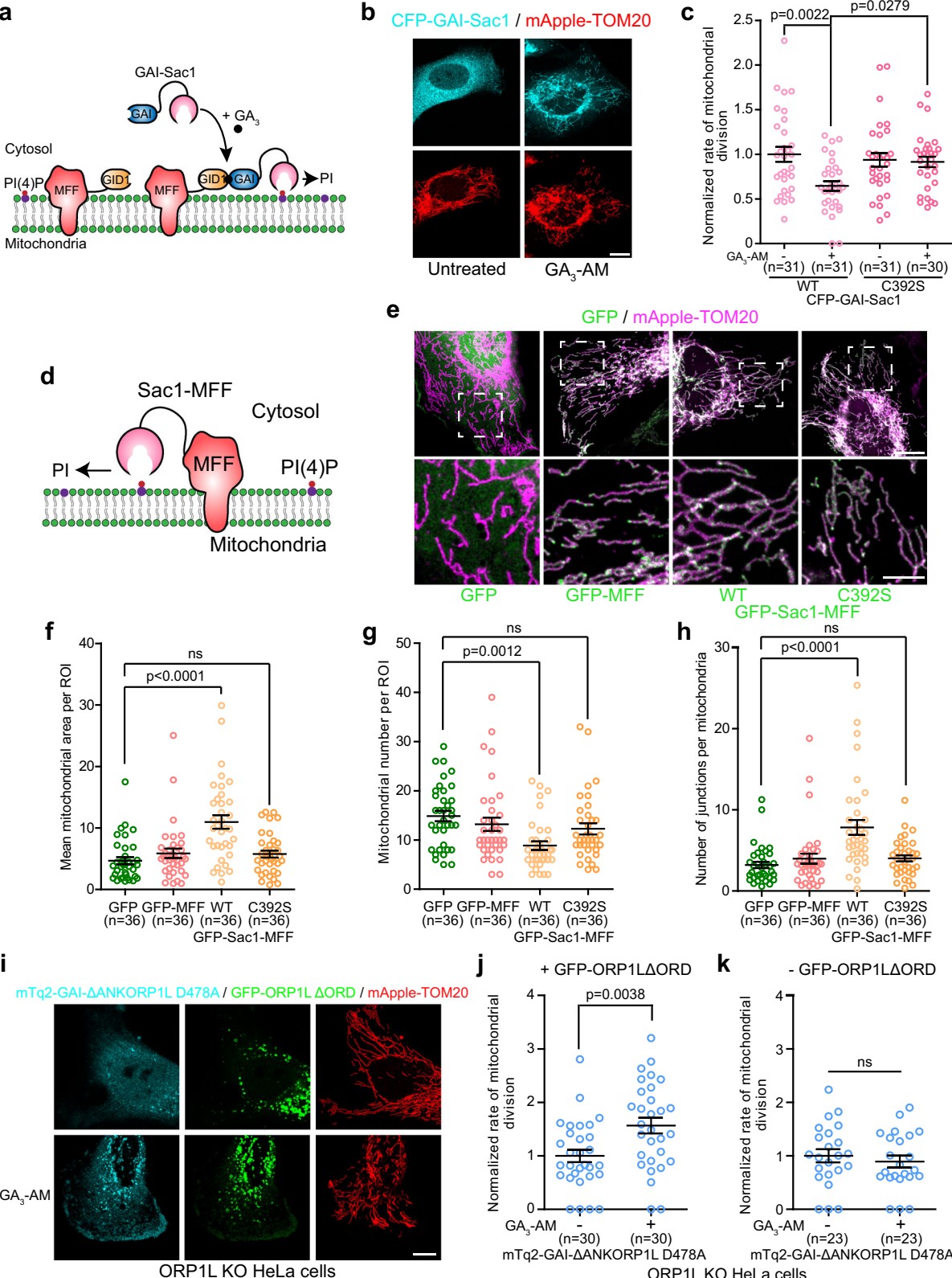

Q67L and 3HA-Rab7 T22N were generated by subcloning Rab7, Rab7 Q67L and Rab7 T22N into pcDNA3.1-3xHA-C1; mApple-Drp1 was made by replacing GFP by mApple (from mApple-TOM20) in the GFP-Drp1 plasmid; CFP-GAI-Sac1 was generated by subcloning Sac1 from GFP-ORPSAC1 into CFP-GAI(1-92) (gift from Takanari Inoue, Addgene # 37307) and CFP-GAI-Sac1 C392S was generated from CFP-GAI-Sac1 via site directed mutagenesis; GFP-ORPOSBP was made by replacing amino acid 334 to 950 of ORP1L from GFP-ORP1L by amino acid 184 to 809 of OSBP from mCherry-OSBP (gift from John Brumell, Hospital for Sick

Children, Toronto); GFP-ORPOSBP HH/AA was made from GFP-ORPOSBP using site directed mutagenesis; iRFP-GID1-MFF was made by replacing Rab7 from iRFP-GID1-Rab7 by MFF from GFP-MFF (gift from Glia Voeltz, Addgene #49153); GFP-Sac1-MFF and GFP-Sac1 C392S-MFF were made by introducing wild-type and C392S mutant Sac1, from CFP-GAI-Sac1 and CFP-GAI-Sac1 C392S respectively, between GFP and MFF in the GFP-MFF plasmid. CFP-GAI-PI4KAc1001 WT and D1957A were made by subcloning PI4KAc1001 WT and D1957A from mCherry-FKBP-PI4KAc1001 WT and D1957A (gift from Gerry

**Fig. 7 The recruitment of the PI(4)P phosphatase Sac1 to mitochondria impairs their division. a** Soluble GAI-Sac1 can be recruited to mitochondrial fission site using GID1-MFF upon GA$_3$-AM treatment, leading to dephosphorylation of PI(4)P at the mitochondrial division site. **b** Representative image of a HeLa cell expressing CFP-GAI-Sac1, iRFP-GID1-MFF (not imaged) and mApple-TOM20 before and after GA$_3$-AM treatment (10 µM). Scale bar: 10 µm. **c** Normalized rate of mitochondrial division before and after GA$_3$-AM (10 µM) treatment in HeLa cells overexpressing the wild-type CFP-GAI-Sac1 or the inactive C392S mutant, iRFP-GID1-MFF and mApple-TOM20. When treated with GA$_3$-AM (10 µM) cells were imaged between 5 and 25 min of treatment. Cells from three independent experiments. Two-way ANOVA, Sidak's multiple comparisons test. **d** In GFP-Sac1-MFF, Sac1 was fused to MFF to anchor it directly to the outer mitochondrial membrane leading to dephosphorylation of PI(4)P at the mitochondrial division site. **e** Representative maximum projection images of HeLa cells expressing GFP, GFP-MFF, GFP-Sac1-MFF or the catalytic inactive GFP-Sac1 C292S-MFF and mApple-TOM20. Inset shows the morphology of the mitochondrial network. Scale bars: 10 µm and 5 µm (inset). **f–h** Mitochondrial morphology was quantified for **f** mean area per mitochondrion, **g** mitochondrial number per region of interest (ROI), and **h** number of junctions per mitochondria. Cells from three independent experiments. One-way ANOVA with Tukey's Multiple Comparison Test. **i** Representative images of ORP1L KO HeLa cells expressing the indicated constructs before and after GA$_3$-AM (10 µM) treatment. Scale bar: 10 µm. **j** Normalized rate of mitochondrial division in ORP1L KO HeLa cells expressing the cytosolic mTq$^2$-GAI-ΔANKORP1L D478A, mApple-TOM20, GFP-ORP1LΔORD, and iRFP-GID1-Rab7 before and after GA$_3$-AM (10 µM) treatment. Cells from three independent experiments. Two-sided unpaired t-test. **k** Normalized rate of mitochondrial division in ORP1L KO HeLa cells expressing the cytosolic mTq$^2$-GAI-ΔANKORP1L D478A, mApple-TOM20 and iRFP-GID1-Rab7 before and after GA$_3$-AM (10 µM) treatment. Cells from three independent experiments. Two-sided unpaired t-test. **c, f–h, j, k** All graphs show the mean ± SEM. ns non-significant: **f** p = 0.7632, **g** p = 0.3592, **h** p = 0.7935, and **k** p = 0.5192.

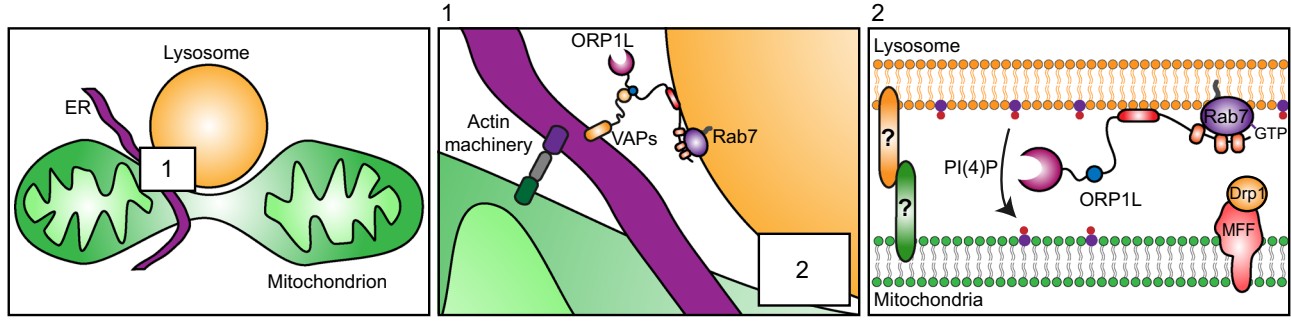

Formation of three-way contact PI(4)P transfer at division site

**Fig. 8 Proposed model for ORP1L functions at the mitochondrial division site.** Mitochondrial division initiates at contacts with the ER, where the ER drives the constriction of mitochondrial membranes. This implicates an actin machinery that involves the ER protein INF2 and the mitochondrial Spire1C. The constriction allows the recruitment of Drp1 by adapters such as MFF and its oligomerization that further constrict mitochondrial membranes (non-represented in the cartoon). Lysosomes are then recruited to the division site in a process mediated by the Rab7-ORP1L-VAPs interaction that establishs contact sites with the ER (**1**). This allows the formation of a three-way contact between the ER, the lysosome and mitochondria at the division site that brings the lysosome in contact with mitochondria. Lysosome-mitochondria tethers are still unknown. At the Lysosome-mitochondria contact (**2**), we propose that ORP1L mediates the transfer of PI(4)P from lysosome to the division site. This model is supported by the impaired mitochondrial division when this transfer is inhibited or when PI(4)P is depleted at the mitochondrial division site.

Hammond, Addgene #139311 and #139312) in CFP-GAI(1-92). MAPL-YFP was a gift from Heidi McBride (McGill University). A complete list of primers used in this study can be found in the Supplementary Table 1. DNA sequences were verified by Sanger sequencing.

**siRNAs**. The following siRNAs, custom synthesized from MilliporeSigma, were used: Rab7 (5′ CUGAACCUAUCAAACUGGA 3′ (ref. [49]), ORP1 (siORP1-1: 5′ GGCAUCAAGAAACACAGAACAA 3′; siORP1-2: 5′ UGAACACAAUGAGGCA UAUAC 3′), VAPA (5′ GCGAAAUCCAUCGAUAGAAA 3′, [50]), VAPB (5′ GCUC UUGGGCUCUGGUGGGUUUU 3′, [50]), STARD3 (5′ GACCUGGUUCCUUGACUU CAA 3′), NPC1 (5′ ACCAAUUGUGAUAGCAAUAUU 3′, [25]), PI4K2B (siPI4K2B-1: 5′ GCUGCAAUUGAUAAUGGUCUA 3′; siPI4K2B-2: 5′ GACAU-GAACUUUGUGCAAGAU 3′), and non-targeting control siRNA (5′-AAUA AGGCUAUGAAGAGAUAC-3′ (ref. [51])).

**Cell culture and transfection**. COS-7, HeLa and U-2 OS cells were obtained from ATCC and were cultured in DMEM (Gibco) supplemented with 10% FBS (Wisent). ORP1L KO HeLa cells generated using CRISPR-Cas9 technology and the ORP1L WT control[19] were a gift from Neale Ridgway (Dalhousie University, Halifax, Canada) and were cultured in the same condition as regular HeLa cells. All cells were cultured at 37 °C in humidified air containing 5% CO2. Cells were transfected using the Neon transfection system (Invitrogen) according to the manufacturer's instructions, using the following parameters: 1050 V, 30 ms and two pulses (COS-7); 1005 V, 35 ms and two pulses (HeLa) ;1230 V, and 10 ms and four pulses (U-2 OS). For overexpression experiments, 5 µg of DNA for 0.5 × 10$^6$ cells were used and cells were imaged or treated 24 h after transfection. For silencing experiments, a siRNA final concentration of 100 nM was used and cells were imaged or treated 48 h after transfection.

**Immunofluorescence**. For immunofluorescence, cells were grown on glass coverslips (#1.5 thickness) and were fixed with PBS containing 4% paraformaldehyde (Electron Microscopy Sciences). Cells were then incubated 10 min at room temperature with PBS containing 100 mM NH$_4$Cl to quench autofluorescence and were subsequently blocked and permeabilized with a solution of 10% FBS, 0.1% Triton-X 100 in PBS for 30 min at room temperature and then incubated with primary antibodies overnight at 4 °C in 10% FBS in PBS. After washing, the cells were incubated with the secondary antibodies for 45 min at room temperature and coverslips were then mounted on glass slides using DAKO (Agilent) mounting medium and stored at 4 °C until imaging. Confocal fluorescence imaging was performed using a Zeiss LSM710 laser-scanning confocal microscope with a 63×1.4 NA Oil Plan-Apochromat objective and the appropriate lasers and filter.

**Live imaging**. Unless indicated all images were acquired on a Leica SP8 Lighting Confocal microscope. For live imaging experiments, cells were grown in Lab-Tek$^{TM}$ chambers (Nunc) with a borosilicate glass bottom. Media was replaced 30 min before imaging and cells were imaged using a Leica SP8 with a 63x glycerol immersion objective lens, 1.3 NA (Leica) in a temperature (37 °C), and atmosphere (5% CO2) controlled environment. Depending on the experiments, single optical section or a Z-stack of 3–5 steps with a step size ranging from 250 nm to 340 nm were acquired. Acquisition was done using the Las X software (Leica) and a deconvolution was performed on every image with the lightning module (Leica) using adaptative parameters. For lysosomes-mitochondria contacts and mitochondrial division experiments; a video of 1 or 2 min with 1 frame every 5 s was acquired. When the presence of the endoplasmic reticulum (ER) at contacts between lysosomes and mitochondria was investigated, or when contacts between lysosomes and the ER were investigated, a 15–30 s video was acquired. Cells imaged

were of similar low intensity in order to be able to compare cells with similar levels of overexpression.

**Gibberellin-induced dimerization system.** For Gibberellin-induced dimerization experiments acquisition GID1 tagged membrane anchor proteins were used to recruit GAI fused cytosolic construct upon $GA_3$-AM treatment (10 μM). Cells were imaged between 5 to 25 min of $GA_3$-AM treatment. Acquisition was performed at 37 °C and 5% CO2 except when the ER fluorescence at lysosome-mitochondria contacts was measured where the treatment and acquisition were done at room temperature in a non-controlled environment. For this experiment $GA_3$-AM was added in the media 30 s after the beginning of the acquisition and 1 frame every minute was acquired for a total of 10 min.

**Structured illumination microscopy.** Samples imaged with structured illumination microscopy (SIM) microscopy were grown on glass coverslips (#1.5 thickness) and were fixed with PBS containing 4% paraformaldehyde (Electron Microscopy Sciences). Imaging of samples was performed on Elyra PS.1 super-resolution inverted microscope (ZEISS) and was done as previously described[50].

**Three-dimensional reconstruction.** For three-dimensional reconstruction of a 3-way contact between lysosome, mitochondria and the ER: COS-7 cells expressing mito-BFP (mitochondria), GFP-KDEL (ER) and Lamp1-mCherry (Lysosomes) were imaged and a three-dimensional reconstruction of the image was performed using Imaris Bitplane (9.2.1) using the "Make surface" feature using default parameters.

**Image analysis.** The proportion of 3-way contacts between the ER, lysosomes and mitochondria were evaluated by quantifying the proportion of lysosome-mitochondria contacts marked by the ER using 3D confocal images from cells expressing mitochondrial, ER and lysosomal markers (Lamp1, Rab7, and ORP1L were used in the study). Contacts between mitochondria and lysosomes, defined as lysosomes in direct association with mitochondria, were identified and the presence of the ER at the site of interaction was then investigated. To avoid potential false negative ER presence at contacts between mitochondria and lysosomes, the analysis was performed in the periphery of the cells where the ER is tubular. The results were expressed as percentage of lysosome-mitochondria contacts marked by the ER.

For the Gibberellin-induced dimerization experiments, the presence of the ER at contacts between lysosomes and mitochondria was measured using ImageJ. In brief, cells were transfected with mitochondrial, ER and lysosomal (iRFP-GID1-Rab7) markers and the fluorescence intensity of the ER marker was measured at contacts between mitochondrial marker and Rab7. First, we created a mask corresponding to Rab7 positive lysosomes using the automatic threshold in ImageJ. A second mask was created for the mitochondrial marker and these two masks were combined to generate a third one corresponding to the pixels shared by the first two masks and, thus, representing lysosome-mitochondria contacts. We quantified the total intensity of ER marker fluorescence in the lysosome-mitochondria contacts mask and expressed it as the percentage of total ER marker fluorescence in every cell for every time-point.

The minimum duration of contacts as well as the percentage of lysosomes (Lamp1-mCherry or mCherry-ORP1L positive vesicles) in contacts with mitochondria were analyzed as previously described[7]. Briefly, 8–12 random contacts (5 for GFP-ORPSAC1) per cell were analyzed. The contacts analyzed were those that had already formed at the beginning of the video and that were lasting for at least 3 consecutive frames (>10 s). The minimum duration of contact was quantified as the time before the dissociation of the lysosome-mitochondria contact over a 2 min video. Contacts that lasted during the totality of the video were scored as 2 min contacts. The percentage of lysosomes in contact with mitochondria was quantified as the percentage of lysosomes that formed contacts with mitochondria divided by the total number of lysosomes that was measured using Volocity (Perkin Elmer). The same method was used to quantify the percentage of lysosomes in contact with the ER.

Mitochondrial division (fission) events were defined as events showing a clear division of one mitochondrion into two distinct mitochondria. The number of mitochondrial division events was then normalized by time and volume to calculate a mitochondrial division rate. A similar strategy was used to measure the mitochondrial fusion rate.

The mitochondrial morphology was analyzed as previously described[6] using ImageJ software. In brief, a maximum projection image was generated from Z-stack images. A region of interest (ROI) of 225 μm² was selected and a manual thresholding was performed. The mean area and number of mitochondria within the ROI were measured using the "Analyze Particles" plugin, excluding particles of <0.1 μm². Then, images were processed twice with the "smooth" function of ImageJ and a manual thresholding was done. Images were then binarized ("Make Binary" function) and skeletonized ("Skeletonize" function). Mean number of junctions per mitochondria was then measured using the "Analyze Skeleton" plugin.

To induce Drp1 mediated mitochondrial division, cells were either transfected with MAPL-YFP or treated with Carbonyl cyanide 3-chlorophenylhydrazone (CCCP) (20 μM) and imaged between 25 and 35 min of treatment. DMSO was used as a control. Mitochondrial morphology was then analyzed as described above.

To estimate the levels of PI(4)P at lysosomes the PI(4)P probe 2xP4M was used[29]. A mask of lysosomes (using Lamp1-mCherry, GFP-ORP1L constructs, GAI-Sac1 constructs or GFP-ORPOSBP depending on the experiment) was generated using automatic threshold of ImageJ and the mean fluorescent intensity of 2xP4M was measured. This was normalized to the plasmalemmal levels of 2xP4M.

Whenever possible, we performed a blind analysis of our data. After acquisition, files were anonymized by other researchers not involved in the study prior to analysis.

**Mitotracker staining.** Cells were incubated for 30 min at 37 °C, 5% $CO_2$ with 10 nM MitoTracker™ Deep Red FM in regular media (DMEM, 10% FBS). Cells were then washed twice with regular media and were imaged after an incubation of 15 min at 37 °C, 5% $CO_2$.

**Western blot.** Cells were lysed in 100 mM Tris-HCL, 1% SDS, and protease inhibitor cocktail (BioShop). Lysates were boiled for 15 min and vortexed briefly every 5 min during this incubation. Lysates were then centrifuged at $21,000 \times g$ for 15 min at room temperature and supernatant were collected. Protein concentration was determined using the BCA assay kit (Thermo Scientific). Proteins were visualized on Clinicselect blue x-rayfilm (Carestream) or using ChemiDoc (Bio-Rad Laboratories). Uncropped and unprocessed version of western blots images can be found in the source data file.

**Quantitative PCR and primers.** Total cellular RNAs were isolated using the Monarch® Total RNA Miniprep Kit (New England BioLabs), and cDNAs were subsequently synthesized using the High-Capacity cDNA Reverse Transcription kit (Applied Biosystems). qPCR was performed on a StepOne Real Time PCR System using SYBR green (ThermoFisher scientific), using β-actin as a reference gene for all quantifications. The list of primers used in this study for qPCR can be found in Supplementary Table 2.

**Graphing and figure assembly.** Graphics were prepared using GraphPad Prism 5. Brightness and contrast of microscopy images were adjusted using ImageJ software for presentation purposes. For experiments with Arf1-GFP (Fig. 1j–l), GFP-Drp1 (Fig. 1g–i), and mApple-Drp1 (Supplementary Fig. 6f), as these constructs exhibit a strong cytosolic signal, brightness and contrast were adjusted to mask the cytosolic signal allowing a better visualization of Arf1 vesicles and Drp1 puncta, for presentation purposes. All final figures were assembled using Illustrator CS6 (Adobe).

**Statistics and reproducibility.** All statistical tests were performed using GraphPad Prism 5 and 9 and are described in the figure legends and provided in full detail in the source data file. A p-value of $P < 0.05$ was considered to be statistically significant. In Supplementary Fig. 5b a Grubbs' test (GraphPad) was performed which resulted in the removal of one outlier from the HH/AA condition. During this study, cells were acquired from three independent experiments, except for Figs. 1g, h, j, k, 2c, 3f, g, 4d, 5a and Supplementary Figs. 1c, 2d–l, k, l, 5e that were from two independent experiments. The number of independent experiments performed is detailed in the figure's legends. For representative images see: Figs. 1a, 2b, e, 3a, b, 4m, 6b, c, 7b, i and Supplementary Figs. 1a, b, 2b–e, i, 6e, f are representative of two independent experiments, Fig. 1b is a representative image of quantification shown in Fig. 1c, d is a representative image of quantification shown in Fig. 1e, g is a representative image of quantification shown in Fig. 1h, j is a representative image of quantification shown in Fig. 1j, Fig. 3c is a representative image of quantification shown in Fig. 3d, e, l is a representative image of quantification shown in Fig. 3m, Figure 4c is a representative image of quantification shown in Fig. 4d, e is a representative image of quantification shown in Fig. 4f–h, Figure 5a is a representative image of quantification shown in Fig. 5b, d is a representative image of quantification shown in Fig. 5e–g, j is a representative image of quantification shown in Fig. 5k, Fig. 6d is a representative image of quantification shown in Fig. 6e, f is a representative image of quantification shown in Fig. 6g–i, Fig. 7e is a representative image of quantification shown in Fig. 7f–h, Supplementary Fig. 2j is a representative image of quantification shown in Fig. 3n, Supplementary Fig. 3a is a representative image of quantification shown in Supplementary Fig. 3b–d, Supplementary Fig. 3a is a representative image of quantification shown in Supplementary Figs. 3b–d and 4a, e are representative images of quantification shown in Supplementary Fig. 4b–d, f–h (respectively), Supplementary Figure 5b is a representative image of quantification shown in Supplementary Fig. 5c, d is a representative image of quantification shown in Supplementary Figure 5e, g is a representative image of quantification shown in Supplementary Figure 5h–j, Supplementary Figure 6a is a representative image of quantification shown in Supplementary Fig. 6b, Supplementary Fig. 7c is a representative image of quantification shown in Supplementary Fig. 7d–f, Supplementary Fig. 8a is a representative image of quantification shown in Supplementary Fig. 8b–d and Supplementary Fig. 8f is a representative image of quantification shown in Supplementary Fig. 8g.

**Reporting summary**. Further information on research design is available in the Nature Research Reporting Summary linked to this article.

## Data availability

All data that support the findings of this study are included in the manuscript or are available from the authors upon reasonable request. Source data are provided with this paper.

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

## Acknowledgements

We thank Nicholas Demers, Laura DiGiovanni, Sharon Leung, and Dr. William Trimble for critically reading the manuscript; Drs. Sergio Grinstein and John Brumell for providing reagents, Dr. Neil Ridgway for providing ORP1L KO HeLa cells; Drs. Kimberly Lau and Paul Paroutis at the SickKids Imaging Facility for assistance with live-cell imaging. Infrastructure for the Kim Laboratory was provided by a John Evans Leadership Fund grant from the Canadian Foundation for Innovation and the Ontario Innovation Trust. This work was supported by operating grants from the Canadian Institutes of Health Research (PJT#156196) to P.K.K.; M.B is a recipient of a Restracomp Fellowship from the Hospital for Sick Children.

## Author contributions

M.B. and P.K.K. conceived and designed the experiments. M.B. performed the experiments and analyzed the data. M.B. and P.K.K. interpreted the data and wrote the paper.

## Competing interests

The authors declare no competing interests.
