## [Peer Review File · Nature Communications]

ORP1L mediated PI(4)P signaling at ER-lysosome-mitochondrion three-way contact contributes to mitochondrial divisionREVIEWER COMMENTS

Reviewer #1 (Remarks to the Author):

Overexpression or knockdown of ORP1L alters three-way contact formation and mitochondrial division. Evidence is presented that ORP1L transports PI(4)P from lysosomes to mitochondria and that PI(4)P on mitochondria is necessary for mitochondrial division. These are exciting findings that suggest an important new role for lysosomes in regulating mitochondrial division, perhaps by providing PI(4)P to mitochondria. While the work is well-done and largely convincing, additional work is necessary to support the claims that PI(4)P plays a role in mitochondrial division and that ORP1L transfers PI(4)P to mitochondria. Alternatively, other models of the role of PI4P and ORP1L in mitochondrial division should be discussed.

1. The study nicely shows that lysosomal PI(4)P is necessary for efficient mitochondrial division but it is less clear that PI(4)P is being transferred to mitochondria. To determine whether PI(4)P on the mitochondria plays a role in mitochondrial division, the authors recruit the PI(4)P phosphatase Sac1 to the surface of mitochondria and find it inhibits division. This is a clever approach, but it remains possible that a small fraction of Sac1 is not on mitochondria or that Sac1 on mitochondria works in trans, hydrolyzing PI(4)P in nearby lysosome (or other organelles). To obtain more evidence for the model, it would be good if the same approach were used to recruit a PI 4-kinase to the surface of mitochondria. If the model is correct, this should bypass the role of ORP1L and perhaps even lysosomes in mitochondrial division. Otherwise, alternative models should be discussed.

2. There is no direct evidence that ORP1L transports PI(4)P from lysosomes to mitochondria (or anywhere else). Indeed, it is hard to see how ORP1L could move PI(4)P from lysosomes to mitochondria when it is bound to VAP on the ER since this would require the three organelles to be tremendously close. It is possible that ORP1L transports PI(4)P when it is not bound to VAP, but that seems inconsistent with the finding that an ORP1L mutant that cannot bind VAP is not functional. Without direct evidence that PI4P is being transported from lysosome to mitochondria, alternative models should be discussed. For example, it remains possible that ORP1L regulates PI(4)P (or cholesterol) pools on lysosomes or the ER that somehow affect mitochondrial division without requiring lipid transport to mitochondria.

Reviewer #2 (Remarks to the Author):

In this manuscript, authors found that lysosomes could be recruited to the mitochondrial division sites by the ER through the Rab7-ORP1L-VAPs interaction at the late step of mitochondrial division. They also demonstrated the protein ORP1L can transport PI(4)P from lysosomes to mitochondrial division sites, while inhibiting this transfer process or depleting PI(4)P impairs mitochondrial fission. These results may help us to get a better understanding of mitochondrial division. Before publication, authors have to address several major issues listed below.

Major issues:

1. In the Fig. 1F, the data cannot demonstrate ER-mito contacts would be preceded by lyso-mito contacts during division. In a normal cell, the number of lysosomes is limited. Therefore, the possibility for ER and mitochondria contact would be high. Authors should find a new parameter to demonstrate this point.
2. In the Fig. 1I, the percentage of sequence is very different to that is shown the Fig. 1F. Authors should give an explanation for this difference.
3. In most figures, the statistical charts are percentages of lysosome-mitochondria contacts marked

by ER. This could be misleading implying that ER is recruited by lyso-mito. Authors should change to percentages of ER-mitochondria contacts marked by lysosomes.

4. One cannot conclude an increase in frequency of mitochondria-lysosome contact (MLC) from the data shown in the Fig. 3D. A higher proportion could be caused by a longer duration shown in Fig 3E. Therefore, since authors didn't directly show any data to support the increase in frequency, the last sentence in the second paragraph of Discussion would be wrong.

5. In Figs. 3D and 3E, ORP1L overexpression could increase MLC proportion and duration, while Figs. 3F and 3G show that ORP1L mediates MLC in an ER dependent manner. However, as shown in Figs. S2G and S2H, ORP1L knockdown did not affect MLC proportion and duration. Author should put some thoughts on the inconsistency.

6. For the various silenced cells showing mitochondrial hyperfusion, the control groups with stimulated mitochondrial fission induced by the overexpression of mitochondrial-anchored protein ligases or by a carbonyl cyanide m-chlorophenylhydrazone treatment should be included. Meanwhile, in other control groups silencing the key components involved in stress-induced mitochondrial hyperfusion, SLP-2 and the pro-fusion factors Mfn1 and Mfn2 need to be checked.

7. PI(4)P is important for the division site. Could authors try to increase the level of PI(4)P on mitochondria and check the fission?

Minor issues:

1. The subtitle of "The Rab7 interactor ORP1L mediates the formation of three-way contacts between lysosomes, mitochondria and the ER" is misleading. It should be "The Rab7 interactor ORP1L mediates the ER-lysosome contacts at the mitochondrial division site".

2. Authors should give more details for the statistical charts.

Reviewer #3 (Remarks to the Author):

This manuscript explores the function of ER-lysosome-mitochondria contact sites in the regulation of mitochondrial fission. Using super-resolution microscopy to image three way contact sites between the organelles, the authors find that the ER contacts mitochondria first, then lysosomes are recruited to the site of division after initial constriction. Overexpression of Rab7 slightly increases the occurrence of these contacts, while ablating Rab7 localization to lysosomes or knockdown by siRNA results in a decrease in these contacts. The authors find that ORP1L is partially required for this three-way contact and that it plays an important role in promoting mitochondrial division, likely via the transport of PI4P from lysosomes to mitochondria). Finally, the authors use artificial tethering and chimeric proteins to further probe the requirement of PI4P in mitochondrial division. In the end, the authors conclude that the ER recruits lysosomes, which supplies PI4P required for mitochondrial fission, to these three-way contacts. Overall, this is an excellent study of very high quality. The imaging is beautiful, and the experiments are nicely laid out and logical. The results suggest an important role for three way mito-lyso-ER contact in mitochondrial fission, and advance our understanding of how organelle contact sites influence organelle behavior. I only have one major comment (#1 below) the authors should address, and several minor comments (#2-6) to consider as well.

1. The only major issue the authors should address with textual changes is the conclusion that Rab7 and ORP1L are required for three-way contacts. While I think the data clearly demonstrates that these proteins are localized to these sites and important for mitochondrial fission, the effect of loss or overexpression of these factors on the formation of three way contacts significant but not extremely robust. As the authors raise in the discussion, it seems likely that either there are other players involved in the formation of these sites, or that the reduction that is seen in these sites in the absence

of these proteins is due a loss of just a subset of three way contacts that are controlled by these factors. I suggest that the authors should modify the text in the results and discussion sections (including the result subheadings) to make it clear that they are only partially required for contact site formation.

2. The authors should better clarify their findings in Supp 1C versus Fig 1E. In Supp 1C, they state that only 3% of 3-way contacts result in mitochondrial division, but in Fig 1E they quantify how many division events include the presence of ER and lysosomes. The difference here is subtle and should be explained in more detail to help the reader.

3. In Figure 3D and E, Lamp1 and ORP1L are overexpressed. A blot to compare the level of over expression might be helpful here. Also, 3D and 3F appear to quantify the same thing/something similar, but WT ORP1L shows a different percentage between the two. Perhaps LAMP1 is a better control for 3F and G, or the differences should be described in the text.

4. It is hard to compare 3D-I, especially when some of the actual side-by-side comparison is in Supp 2 F-H. The authors should show more consistent analyses in the main figures.

5. How Fig 4A was quantified should be described in more detail. Especially because Supp 3B and C show no significant change for the Q67L mutant.

6. Supp 3F is hard to see. It would be helpful to show a different representative blot.

We thank all the reviewers for their helpful comments. We feel that addressing their concerns improved both the readability and substance of the manuscript. We have included couple of new experiments to aid in the interpretation of our findings. We have also made changes to the manuscript to address the reviewers concerns and also to improve the readability of the manuscript. New texts are shown in RED in the manuscript.

Reviewer #1 (Remarks to the Author)

Overexpression or knockdown of ORP1L alters three-way contact formation and mitochondrial division. Evidence is presented that ORP1L transports PI(4)P from lysosomes to mitochondria and that PI(4)P on mitochondria is necessary for mitochondrial division. These are exciting findings that suggest an important new role for lysosomes in regulating mitochondria division, perhaps by providing PI(4)P to mitochondria. While the work is well-done and largely convincing, additional work is necessary to support the claims that PI(4)P plays a role in mitochondrial division and that ORP1L transfers PI(4)P to mitochondria. Alternatively, other models of the role of PI4P and ORP1L in mitochondrial division should be discussed.

We thank the reviewer for the supportive evaluation of our work. As pointed out by the reviewer others model are possible and we added some text at the end of the discussion to acknowledge this possibility. Additionally, we performed additional experiments suggested by the reviewer.

1. The study nicely shows that lysosomal PI(4)P is necessary for efficient mitochondrial division but it is less clear that PI(4)P is being transferred to mitochondria. To determine whether PI(4)P on the mitochondria plays a role in mitochondrial division, the authors recruit the PI(4)P phosphatase Sac1 to the surface of mitochondria and find it inhibits division. This is a clever approach, but it remains possible that a small fraction of Sac1 is not on mitochondria or that Sac1 on mitochondria works in trans, hydrolyzing PI(4)P in nearby lysosome (or other organelles). To obtain more evidence for the model, it would be good if the same approach were used to recruit a PI 4-kinase to the surface of mitochondria. If the model is correct, this should bypass the role of ORP1L and perhaps even lysosomes in mitochondrial division. Otherwise, alternative models should be discussed.

Response: As suggested by the reviewer, we targeted a PI 4-Kinase catalytic domain using the GAI-GID1 system to mitochondria and tested whether it affects the mitochondrial division rate. We used the catalytic domain of PI4KA (PI4KAc1001) as this PI4K catalytic fragment was successfully recruited to mitochondria to generate PI(4)P (Zewe et al., 2020). Targeting PI4Kc1001 to mitochondria increased PI(4)P on mitochondria, but it did not affect the mitochondrial division rate in both control HeLa or ORP1L KO cells suggesting that the production of PI(4)P alone on mitochondria does not promote fission or bypass the need for ORP1L/lysosomes. We attached the new figure below and placed the new experiment in **Supplementary Figure 8** (and lines 345-353)

e Soluble GAI-PI4KA₁₀₀₁ (catalytic domain of PI4KA) can be recruited to mitochondrial fission site using GID1-MFF upon GA_3 -AM treatment, leading to production of PI(4)P at the mitochondrial division site. **f, g** (f) Representative images of HeLa cells expressing CFP-GAI-PI4KA₁₀₀₁ or the catalytic dead D1957A mutant, iRFP-GID1-Rab7 and mCherry-2xP4AM after GA_3 -AM treatment. Scale bars: 10 μ m and 1 μ m (inset). **(g)** Quantification of the mitochondrial levels, normalized by the plasmalemmal levels, of 2xP4AM. The graphs show the mean \pm SEM, n>29 cells per condition analyzed in three independent experiments. Unpaired t-test. *p<0.05. **h, i** Normalized rate of mitochondrial division before and after GA_3 -AM (10 μ M) treatment in **(h)** HeLa cells or **(i)** ORP1L KO HeLa cells overexpressing the wild-type CFP-GAI-PI4KA₁₀₀₁ or the inactive D1957A mutant, iRFP-GID1-MFF and mApple-TOM20. When treated with GA_3 -AM (10 μ M) cells were imaged between 5 and 25 minutes of treatment. **(h)** n>30 cells and **(i)** n>24 cells per condition analyzed in three independent experiments. Two-way ANOVA, Sidak's multiple comparisons test, ns = non-significant. The graphs show the mean \pm SEM.

PI(4)P, like other phosphorylated inositols, are potent signaling molecules and act in both the recruitment and regulation of various proteins. And recent work in endocytosis has shown that PI4P levels are rapidly modified for inducing morphological and function changes in organelles. Therefore, it is not surprising that constitutive synthesis of PI(4)P on mitochondria did not affect the mitochondrial fission rate. We interpret these results to suggest that either PI(4)P at the mitochondria is not the rate-limiting step and/or lysosomes/ORP1L are probably also regulating others factors during the division process. We discussed some of the roles of the ORP1L PI(4)P transport function, and alternative models, in the discussion section (lines 439-448).

2. There is no direct evidence that ORP1L transports PI(4)P from lysosomes to mitochondria (or anywhere else). Indeed, it is hard to see how ORP1L could move PI(4)P from lysosomes to mitochondria when it is bound to VAP on the ER since this would require the three organelles to be tremendously close. It is possible that ORP1L transports PI(4)P when it is not bound to VAP, but that seems inconsistent with the finding that an ORP1L mutant that cannot bind VAP is not functional. Without direct evidence that PI4P is being transported from lysosome to mitochondria, alternative models should be discussed. For example, it remains possible that ORP1L regulates PI(4)P (or

cholesterol) pools on lysosomes or the ER that somehow affect mitochondrial division without requiring lipid transport to mitochondria.

Response: The reviewer is correct that in our model of ORP1L dependent PI(4)P transport that it must be unbound to VAPs if its to transport PI(4)P to mitochondria. This model is possible given that Rab7 has been implicated in lysosome-mitochondria contact sites (Wong 2018). For this reason, we propose that the ORP1L FFAT motif and ORD domain may not be directly dependent on each other and that ORP1L could transfer lipids without binding to VAPs. Therefore, we propose that ORP1L has two different functions that act sequentially for mitochondrial division: 1) the formation of the ER-lysosome-mitochondria three-way contact and 2) PI(4)P transfer at lysosome-mitochondria contact. The first step requires the FFAT motif as the ER recruits lysosomes to the site of division (involving the Rab7-ORP1L-VAPs interaction). For lipid transport function, we argue that it does not require ORP1L to bind to VAP. To support this model, we performed an additional experiment where we separated the two functions by expressing the ORP1L Δ ORD constructs that can bind to VAPs (and thus promote three-way contacts formation) but do not possess the lipid transfer domain; and the FFAT mutant/ANK deleted construct Δ ANKORP1L D478A. We used the GAI-GID1 dimerization system to recruit the cytosolic Δ ANKORP1L D478A that possess the lipid transfer domain (ORD) but cannot bind to VAPs and thus cannot promote three-way contacts formation (Fig. 3n). This setup allowed us to segregate the two functions of ORP1L in the division process. We found that Δ ANKORP1L D478A recruitment to lysosomes immediately increased the mitochondrial division rate in these cells only when co-expressed with ORP1L Δ ORD. This result suggests that ORP1L does not need to bind to VAPs to transfer lipid at the three-way contact sites to rescue mitochondria fission defect in the ORP1L KO cells. These results further support ORP1L transferring PI(4)P at the mitochondria-lysosome contact site (see proposed model in Fig. 8). We placed this new data in Fig. 7i-k and in lines 354-372. Additionally, as suggested by the reviewer we discussed alternative models (lines 444-448).

i Representative images of ORP1L KO HeLa cells expressing the indicated constructs before and after GA₃-AM (10μM) treatment. j

Normalized rate of mitochondrial division in ORP1L KO HeLa cells expressing the cytosolic mTq²-GAI- Δ ANKORP1L D478A, mApple-TOM20, GFP-ORP1L Δ ORD and iRFP-GID1-Rab7 before and after GA₃-AM (10μM) treatment. n= 30 cells per condition

analyzed in three independent experiments. Unpaired t-test. k Normalized rate of mitochondrial division in ORP1L KO HeLa cells expressing the cytosolic mTq²-GAI- Δ ANKORP1L D478A, mApple-TOM20 and iRFP-GID1-Rab7 before and after GA₃-AM (10μM) treatment. n= 23 cells per condition analyzed in three independent experiments. Unpaired t-test. All graphs show the mean \pm SEM.

*** p<0.001, **p<0.01, *p<0.05 and ns = non-significant.

Reviewer #2 (Remarks to the Author):

In this manuscript, authors found that lysosomes could be recruited to the mitochondrial division sites by the ER through the Rab7-ORP1L-VAPs interaction at the late step of mitochondrial division. They also demonstrated the protein ORP1L can transport PI(4)P from lysosomes to mitochondrial division sites, while inhibiting this transfer process or depleting PI(4)P impairs mitochondrial fission. These results may help us to get a better understanding of mitochondrial division. Before publication, authors have to address several major issues listed below.

Major issues:

1. In the Fig. 1F, the data cannot demonstrate ER-mito contacts would be preceded by lyso-mito contacts during division. In a normal cell, the number of lysosomes is limited. Therefore, the possibility for ER and mitochondria contact would be high. Authors should find a new parameter to demonstrate this point.

Response: We agree that the data in **Fig. 1f** comparing the quantification of ER-mitochondria contacts with respect to lysosome-mitochondria contact preceding mitochondria division may be skewed towards ER-mitochondria contacts due to the limited number of lysosomes. However, our conclusion that the ER precedes lysosomes to the division site is not based on the data presented in **Fig. 1f** alone. First, we show that 100% of all mitochondria division sites have ER juxtaposition, while only about 60% have lysosomes (**Fig 1e**). Given that all fission events were marked by ER, we do not believe this to be a random event. This is also consistent with several other studies supporting that the mitochondrial division process initiates at ER-mitochondria contacts, where the ER mediates a pre-constriction of mitochondrial membranes (Friedman et al, 2011; Kraus and Ryan, 2017). Further, we found that the vast majority of lysosomes are recruited after Drp1 oligomerization on mitochondria (**Fig. 1i**). Since Drp1 is recruited to the division site after pre-constriction of mitochondrial membranes by the ER (Kraus and Ryan, 2017), our data strongly suggest lysosomes are recruited after the ER.

2. In the Fig. 1l, the percentage of sequence is very different to that is shown the Fig. 1F. Authors should give an explanation for this difference.

Response: The difference in the percentage of sequence between ER/lysosomes and Drp1/Lysosomes is due to the time interval between the events. It is known that the ER initiates the division process by pre-constricting the mitochondria which allow for the accumulation of Drp1 at the site of division. Our finding places lysosome recruitment after Drp1 recruitment. Thus, the time interval between Drp1/lysosome recruitment is less than that of ER/lysosomes. The 9% of division events that we observed where lysosome preceded ER is within a reasonable margin of error of the assay and thus suggest that ER precedes lysosomes. However, as the time interval between Drp1 and lysosome is much shorter this decreases the assay's dynamic range. The closer interval can not only increase error, but it is also susceptible to biological variability where lysosome may be recruited before or at the same time of Drp1. Nevertheless, our quantification shows that in majority of fission events the recruitment of Drp1 precedes lysosome recruitment. We did not expand on the explanation of the difference in percentage in the text for the sake of brevity and as we felt the description of the recruitment event was discussed already in the text. However, we added to the text to better explain the significance of **Fig. 1i** (line 112).

3. In most figures, the statistical charts are percentages of lysosome-mitochondria contacts marked

by ER. This could be misleading implying that ER is recruited by lyso-mito. Authors should change to percentages of ER-mitochondria contacts marked by lysosomes.

Response: The problem with using the ER-mitochondria contacts marked by lysosomes is the loss of the assay's dynamic range. As the reviewer pointed out in point #1, the number of contacts between lysosomes and mitochondria is significantly less compared to the number of ER-mitochondria contacts due to the smaller number of lysosomes present in the cell. Therefore, calculating the percentages of ER-mitochondria contact marked by lysosomes will be a very small value, thus decreasing the dynamic range of the assay. Further, the smaller number of lysosome-mitochondria contact sites makes it easier to identify all contacts present at any time point compared to quantifying all the ER-mitochondria contact sites. Thus, using Lysosome-mitochondria contacts first, decreases the potential error due to missed counts.

To help prevent misunderstanding by the readers of the meaning of “percentages of lysosome-mitochondria contacts marked by ER” we clarified the assay in the text (line 84-87). Further, we have also emphasized our interpretation of the data that the ER is recruiting lysosomes to the site of fission in the discussion section (lines 450-452).

4. One cannot conclude an increase in frequency of mitochondria-lysosome contact (MLC) from the data shown in the Fig. 3D. A higher proportion could be caused by a longer duration shown in Fig 3E. Therefore, since authors didn't directly show any data to support the increase in frequency, the last sentence in the second paragraph of Discussion would be wrong.

Response: We agree with the reviewer and decided to delete this sentence from the discussion.

5. In Figs. 3D and 3E, ORP1L overexpression could increase MLC proportion and duration, while Figs. 3F and 3G show that ORP1L mediates MLC in an ER dependent manner. However, as shown in Figs. S2G and S2H, ORP1L knockdown did not affect MLC proportion and duration. Author should put some thoughts on the inconsistency.

Response: As pointed out by the reviewer, the overexpression of ORP1L increased contacts between lysosomes and mitochondria, while the siRNA mediated depletion of ORP1L showed not significant change. This result suggests that ORP1L is sufficient but not necessary to induce mitochondria-lysosome contacts. It shows that overexpressing ORP1L can induce an increase in contact between these two organelles, likely through promoting ER-lysosome-mitochondria three-way contacts. However, the knockdown studies suggest that ORP1L is not the dominant, or direct, tether between mitochondria and lysosomes at basal state. On lines 175-177, we added a sentence suggesting an explanation for the siRNA result.

6. For the various silenced cells showing mitochondrial hyperfusion, the control groups with stimulated mitochondrial fission induced by the overexpression of mitochondrial-anchored protein ligases or by a carbonyl cyanide m-chlorophenylhydrazone treatment should be included. Meanwhile, in other control groups silencing the key components involved in stress-induced mitochondrial hyperfusion, SLP-2 and the pro-fusion factors Mfn1 and Mfn2 need to be checked.

Response: We agree with the reviewer that it is important to check that the hyperfusion was not caused by an upregulation of fusion activity and only due to a decrease mitochondrial division rate. Instead of silencing SLP-2 or Mfn1/2 as controls for fusion, we measured the mitochondrial fusion rate (number of mitochondrial fusion events normalized by time and cell volume) after siRNA depletion of ORP1L that induces a hyperfusion (**Fig. 4e-h**) to account for possible changes in

mitochondrial fusion due to ORP1L depletion. We did not detect any difference in the fusion rate (**Supplementary Fig. 3i**) in these cells compared to cells treated with a control siRNA, indicating that hyperfusion is solely due to decreased fission activity and not to an increased fusion when ORP1L is downregulated. This new data is included in **Supplementary Fig. 3i (also below)** and described on line 220.

i Normalized rate of mitochondrial fusion in HeLa cells treated with the indicated siRNAs and expressing mito-GFP. $n > 28$ cells per condition analyzed in three independent experiments. One-way ANOVA, with Dunnett's Multiple Comparison Test. ns = non statistically significant. The graphs show the mean \pm SEM.

As suggested by the reviewer, we also examined whether silencing ORP1L affected mitochondria morphology in conditions that stimulates Drp1 dependent mitochondrial division by treating cells with CCCP or by overexpressing MAPL after siRNA depletion of ORP1L. We found that siRNA depletion of ORP1L strongly protected mitochondria against the Drp1 dependent stimulated mitochondrial division (see below) supporting an important role of ORP1L in the mitochondrial division process, downstream of Drp1. This data found in **Supplementary Fig. 4a-h** and is described on lines 225-228.

Supplementary Figure 4. Loss of ORP1L protects from Drp1 dependent stimulated mitochondrial division.

a Representative maximum projection images of mitochondrial morphology in HeLa cells overexpressing mApple-TOM20 and treated with the indicated siRNA. Cells were treated with CCCP (20μM) for 30 minutes to stimulate mitochondrial division in a Drp1 dependent manner. Scale bars: 10μm and 5μm (inset). **b-d** Mitochondrial morphology was quantified for **(b)** mean area per mitochondrion, **(c)** mitochondrial number per region of interest (ROI) and **(d)** number of junctions per mitochondria. n=35 cells per condition analyzed in three independent experiments. Two-way ANOVA with Tukey's Multiple Comparison Test, ***p<0.001, **p<0.01, and *p<0.05. The graphs show the mean ± SEM. **e** Representative maximum projection images of mitochondrial morphology in HeLa cells overexpressing GFP or MAPL-YFP (which stimulates Drp1 dependent mitochondrial division) and treated with the indicated siRNA. Mitotracker was used to visualize mitochondria. Scale bars: 10μm and 5μm (inset). **f-h** Mitochondrial morphology was quantified for **(f)** mean area per mitochondrion, **(g)** mitochondrial number per region of interest (ROI) and **(h)** number of junctions per mitochondria. n=40 cells per condition analyzed in three independent experiments. Two-way ANOVA with Tukey's Multiple Comparison Test, ***p<0.001, **p<0.01, and *p<0.05. The graphs show the mean ± SEM.

7. PI(4)P is important for the division site. Could authors try to increase the level of PI(4)P on mitochondria and check the fission?

Response: As suggested by the reviewer, we increased PI(4)P on mitochondria by targeting the catalytic domain of PI4KA (**Supplementary Figure 8- also see our response to reviewer 1 point 1**), but we did not see any effect on the mitochondrial division rate. Our interpretation of this is that PI(4)P by itself cannot promote mitochondrial division and that other factors are probably regulated by lysosomes/ORP1L at these sites. We discuss some of these factors such as ORP1L acting as counter-transport (PI(4)P against cholesterol) in the discussion section (lines 439-444).

Minor issues:

1. The subtitle of "The Rab7 interactor ORP1L mediates the formation of three-way contacts between lysosomes, mitochondria and the ER" is misleading. It should be "The Rab7 interactor ORP1L mediates the ER-lysosome contacts at the mitochondrial division site".

Response: We thank the reviewer for pointing out this. We modified the subtitle to "**ORP1L mediates the Rab7 dependent ER-lysosome-mitochondria contact**" as we felt this better address the main point of the section. In this section we do not yet demonstrate that ORP1L is required for mitochondria fission, but instead show that ORP1L regulates a subset of three-way contacts between the three organelles.

2. Authors should give more details for the statistical charts.

Response: We added the full statistical data (from GraphPad) to the source data file provided with the manuscript along with the Raw data for every experiment.

Reviewer #3 (Remarks to the Author):

This manuscript explores the function of ER-lysosome-mitochondria contact sites in the regulation of mitochondrial fission. Using super-resolution microscopy to image three way contact sites between the organelles, the authors find that the ER contacts mitochondria first, then lysosomes are recruited to the site of division after initial constriction. Overexpression of Rab7 slightly increases the occurrence of these contacts, while ablating Rab7 localization to lysosomes or knockdown by siRNA results in a decrease in these contacts. The authors find that ORP1L is partially required for this three-way contact and that it plays an important role in promoting mitochondrial division, likely via the transport of PI4P from lysosomes to mitochondria). Finally, the authors use artificial tethering and chimeric proteins to further probe the requirement of PI4P in mitochondrial division. In the end, the authors conclude that the ER recruits lysosomes, which supplies PI4P required for mitochondrial fission, to these three-way contacts. Overall, this is an excellent study of very high quality. The imaging is beautiful, and the experiments are nicely laid out and logical. The results suggest an important role for three way mito-lyso-ER contact in mitochondrial fission, and advance our understanding of how organelle contact sites influence organelle behavior. I only have one major comment (#1 below) the authors should address, and several minor comments (#2-6) to consider as well.

We would like to thank the reviewer for this positive evaluation of our manuscript and made several text changes to address the reviewer's concerns.

1. The only major issue the authors should address with textual changes is the conclusion that Rab7 and ORP1L are required for three-way contacts. While I think the data clearly demonstrates that these proteins are localized to these sites and important for mitochondrial fission, the effect of loss or overexpression of these factors on the formation of three way contacts significant but not extremely robust. As the authors raise in the discussion, it seems likely that either there are other players involved in the formation of these sites, or that the reduction that is seen in these sites in the absence of these proteins is due a loss of just a subset of three way contacts that are controlled by these factors. I suggest that the authors should modify the text in the results and discussion sections (including the result subheadings) to make it clear that they are only partially required for contact site formation.

Response: We agree with the reviewer that our data shows that ORP1L mediates a subset of ER-lysosome-mitochondria three-way contacts. We also agree that our data shows that these ORP1L mediated contact sites mediates a subset of mitochondrial fission as we see lysosomes and ORP1L at about 60% of all division site. We have made modifications throughout the text in both Results and discussion sections, including changing of a subtitle to make it clear that Rab7 and ORP1L are involved in a subset of three-way contacts which are involved in a subset of mitochondrial division.

These changes can be seen in:

Line 113: change in subtitle

Lines 127-129: change in sentence stating that Rab7 is involved in a subset of 3-way contacts

Line 171: stating that the reduced expression of ORP1L does not abolish 3-way contact

Lines 175-177: ORP1L is not the dominant tether for lysosome and mitochondria

Line 181 ,198-199: stating that ORP1L is involved in a subset of 3-way contact

Lines 398-400: discussion of Rab7-ORP1L-VAPs forming a subset of 3-way contact

Line 782: change of a title figure to state that Rab7 regulates formation of a subset of three-way contacts.

2. The authors should better clarify their findings in Supp 1C versus Fig 1E. In Supp 1C, they state that only 3% of 3-way contacts result in mitochondrial division, but in Fig 1E they quantify how many division events include the presence of ER and lysosomes. The difference here is subtle and should be explained in more detail to help the reader.

Response: We agree that the subtle difference between **Supplementary Fig. 1c** and **Fig 1e** may be confusing. We have clarified the difference between the two on lines 97-98.

3. In Figure 3D and E, Lamp1 and ORP1L are overexpressed. A blot to compare the level of over expression might be helpful here. Also, 3D and 3F appear to quantify the same thing/something similar, but WT ORP1L shows a different percentage between the two. Perhaps LAMP1 is a better control for 3F and G, or the differences should be described in the text.

Response: As Lamp1-mCherry and ORP1L-mCherry constructs transfect with different efficiency and show different expression within individual cells, western blot is not suitable for analyzing the overexpression level for single cell-based experiments. To mitigate the issue of differential protein levels between cells, we only imaged cells with low and similar levels (by intensity) of fluorophore

intensity. In our experience, fluorophore intensity is an excellent method to compare the levels of proteins that are tagged with the same fluorophore in the same cellular environment. We added some clarification on this subject in the method section (lines 528-529)

The difference between mCherry-ORP1L (**Fig. 3d**) and GFP-ORP1L (**Fig. 3f**) is due to the difference in the type of lysosomes being quantified. In **Fig. 3d** we are comparing ORP1L positive lysosomes vs lamp1 positive lysosomes in contact with mitochondria. This is to address whether ORP1L may increase lysosome-mitochondria contacts. Therefore, in **Fig. 3d** we measured the percentage of mCherry-ORP1L positive lysosomes in contact with mitochondria and compared to mCherry-Lamp1 positive lysosomes. In **Fig. 3f**, we quantified all mCherry-Lamp1 positive lysosomes in cells expressing one of three GFP constructs. The purpose was to compare cells expressing ORP1L vs FFAT mutant ORP1L. Since ORP1L is only found in a subset of lamp1 positive lysosomes (**supplementary fig. 2b**), this results in a difference in percentage of ORP1L positive lysosomes in contact with mitochondria between **Fig 3d** and in Lamp1 positive lysosomes (when ORP1L is overexpressed) in contact with mitochondria in **Fig 3f**. We have made it clear in the manuscript what is being measured for **Fig. 3f** on line 164.

4. It is hard to compare 3D-I, especially when some of the actual side-by-side comparison is in Supp 2 F-H. The authors should show more consistent analyses in the main figures.

Response: We have separated the figures between **Fig. 3** and **Supplementary Fig. 2** for sake of brevity. To help the readers we have made adjustments in how we present the data in the written portion of the manuscript.

5. How Fig 4A was quantified should be described in more detail. Especially because Supp 3B and C show no significant change for the Q67L mutant.

Response: In **Fig. 4a** we quantified the mitochondrial division rate for cells overexpressing the wild-type Rab7 and the Q67L and T22N mutants. The number of mitochondrial division events was normalized by time and volume to calculate a mitochondrial division rate. We added this description in the figure legend to further clarify (“number of mitochondrial divisions normalized by time and volume”). We then analyzed the mitochondrial morphology for these cells and did not detect a significant alteration of the mitochondrial area (only a trend) or number for the Q67L mutant. While the decreased division rate is clear and was reported before (Wong et al, 2018) it is true that the effect on the morphology is subtle. This is perhaps due to an effect of Rab7 Q67L on the mitochondrial fusion process that could affect the morphology. As we have not tested this hypothesis, we feel it is better to not include it in the text. We modified the text on lines 206-207 to indicate clearly that elongation and hyperfusion were observed more robustly on cells expressing the T22N mutant.

6. Supp 3F is hard to see. It would be helpful to show a different representative blot.

Response: We provided a better version of the former blot.

REVIEWERS' COMMENTS

Reviewer #1 (Remarks to the Author):

The revised version of this manuscript is only marginally stronger than the previous one. As I said in my previous review, this study suggests an important new role for lysosomes in regulating mitochondria division, which is a significant discovery. However, there is still no evidence (zero) that PI(4)P is being moved anywhere. The new evidence says nothing about transport; it just shows that all the domains of ORP1L are necessary to support mitochondrial division even when the domains are present on separate proteins. In addition, the new demonstration producing PI(4)P directly on mitochondria does alter the mitochondrial division rate or bypass the role ORP1L in division (new Sup. Fig. 8) is hard to square with the authors' model. They suggest that PI(4)P has to be delivered at the right time, but there is no evidence for that. To this reviewer, the idea that PI(4)P is being moved anywhere is little more than an attractive hypothesis. So why argue so forcefully for the idea that transport is occurring, including putting it in the title? Of course, the authors should suggest any model they like. But I respectfully suggest that the study is just as strong, and it would be better for the field, if the authors substantially toned down their claims about transport and wrote a more balance discussion of their findings. For example, the title could be changed to something like "ORP1L promotes 3-way contacts between the ER, lysosome, and mitochondria that facilitate mitochondrial division."

Reviewer #2 (Remarks to the Author):

All my concerns have been addressed.

Reviewer #3 (Remarks to the Author):

The authors have nicely addressed all of my previous concerns and the work represents an interesting and important step forward in the field.

Reviewer #1 (Remarks to the Author):

The revised version of this manuscript is only marginally stronger than the previous one. As I said in my previous review, this study suggests an important new role for lysosomes in regulating mitochondria division, which is a significant discovery. However, there is still no evidence (zero) that PI(4)P is being moved anywhere. The new evidence says nothing about transport; it just shows that all the domains of ORP1L are necessary to support mitochondrial division even when the domains are present on separate proteins. In addition, the new demonstration producing PI(4)P directly on mitochondria does alter the mitochondrial division rate or bypass the role ORP1L in division (new Sup. Fig. 8) is hard to square with the authors' model. They suggest that PI(4)P has to be delivered at the right time, but there is no evidence for that. To this reviewer, the idea that PI(4)P is being moved anywhere is little more than an attractive hypothesis. So why argue so forcefully for the idea that transport is occurring, including putting it in the title? Of course, the authors should suggest any model they like. But I respectfully suggest that the study is just as strong, and it would be better for the field, if the authors substantially toned down their claims about transport and wrote a more balanced discussion of their findings. For example, the title could be changed to something like "ORP1L promotes 3-way contacts between the ER, lysosome, and mitochondria that facilitate mitochondrial division."

We thank the reviewer for his/her thought-provoking comment and suggestion. Again, we feel that the reviewer's thoughtful comments have helped us improve our manuscript. We agree with the reviewer that our data do not directly prove that a PI(4)P transfer is occurring and that ORP1L could regulate PI(4)P levels without transporting it to mitochondria directly. However, we do provide a significant amount of data that PI(4)P plays a crucial role at ER-lysosome-mitochondria three-way contacts during mitochondrial division, in an ORP1L dependant manner. We also provide a number of indirect evidence to give supports to the model where ORP1L transports PI(4)P to the mitochondrial division site. This includes our observation that mitochondrial division was altered by targeting the PI(4)P phosphatase Sac1 to mitochondrial membranes, which supports the need for PI(4)P at the division site. Further, forced targeting of a bona fide PI(4)P transfer protein, OSBP, to Rab7 lysosomes rescued the mitochondrial morphology in cells depleted of ORP1L, suggesting that a PI(4)P transporter is required. OSBP moves PI(4)P down its concentration gradient to exchange it for moving another lipid (cholesterol) against its concentration gradient. We also demonstrate that the ER tethering and lipid transport function of ORP1L can be decoupled (Fig. 7i-k). Given that Rab7 has been shown to be at the lysosome-mitochondria membrane contact site (MCS), and ORP1L binds to Rab7, our decoupling experiment further supports the model that ORP1L can be localized and act at the Lysosome-Mitochondria MCS. We agree that these are indirect evidence of a transfer. Therefore we, as suggested by the reviewer, toned down our conclusions (changes are highlighted in yellow) and modified the title of the manuscript. However, as we do believe that our study demonstrate that PI(4)P plays a crucial role, we feel it is important to include it in the title. We propose the following new title: "ORP1L mediated PI(4)P signaling at ER-lysosome-mitochondrion three-way contact contributes to mitochondrial division".

Reviewer #2 (Remarks to the Author):

All my concerns have been addressed.

Reviewer #3 (Remarks to the Author):

The authors have nicely addressed all of my previous concerns, and the work represents an interesting and important step forward in the field.

We thank Reviewers #2 and #3 for their insightful comments during the review process.